# Learning to Reason for Hallucination Span Detection

Hsuan Su[♡][*]    Ting-Yao Hu[◇]    Hema Swetha Koppula[◇]    Kundan Krishna[◇]
Hadi Pouransari[◇]    Cheng-Yu Hsieh[◇]    Cem Koc[◇]    Joseph Yitan Cheng[◇]
Oncel Tuzel[◇]    Raviteja Vemulapalli[◇]

[♡]National Taiwan University    [◇]Apple

## Abstract

Large language models (LLMs) often generate hallucinations—unsupported content that undermines reliability. While most prior works frame hallucination detection as a binary task, many real-world applications require identifying hallucinated spans, which is a multi-step decision making process. This naturally raises the question of whether explicit reasoning can help the complex task of detecting hallucination spans. To answer this question, we first evaluate pretrained models with and without Chain-of-Thought (CoT) reasoning, and show that CoT reasoning has the potential to generate at least one correct answer when sampled multiple times. Motivated by this, we propose RL4HS, a reinforcement learning framework that incentivizes reasoning with a span-level reward function. RL4HS builds on Group Relative Policy Optimization and introduces Class-Aware Policy Optimization to mitigate reward imbalance issue. Experiments on the RAGTruth benchmark (summarization, question answering, data-to-text) show that RL4HS surpasses pretrained reasoning models and supervised fine-tuning, demonstrating the necessity of reinforcement learning with span-level rewards for detecting hallucination spans.

## 1    Introduction

Over the past few years, Large Language Models (LLMs) have demonstrated remarkable capabilities across a wide range of natural language processing tasks (Xie et al., 2023; Zhang et al., 2023; Gao et al., 2024; OpenAI et al., 2024). However, they are still prone to generating *hallucinations*—content that is not supported by the input context or the underlying knowledge sources (Zhu et al., 2024; Kalai et al., 2025; Huang et al., 2025). Hallucinations pose critical risks in downstream applications such as summarization and long-form question answering, where reliability and factual consistency with respect to the input context are paramount. Hence, the ability to detect hallucinations is crucial for successful real-world deployment of LLMs.

Most existing research works focus on *binary hallucination detection* problem, where the goal is to determine if the model output contains hallucinations or not (Yang et al., 2024a;b; Tang et al., 2024; Ravi et al., 2024; Ji et al., 2024; Chuang et al., 2024). While useful, this formulation is limited: in many real-world applications, one often needs to know which specific spans in the model output are hallucinated in order to assess the reliability of the generated content. This motivates the problem of *hallucination span detection*, where the goal is to precisely locate unsupported content in the model output (Wu et al., 2023; Ogasa & Arase, 2025).

Recently, *reasoning*—the process of systematically arriving at conclusions by generating and utilizing intermediate steps—has been shown to significantly enhance the capabilities of LLMs in solving complex tasks such as mathematics (Shao et al., 2024; Yu et al., 2025) and coding (Liu & Zhang, 2025; Chen et al., 2025). Hallucination span detection is also a complex multi-step decision making process as it requires carefully analyzing the model output to extract all the stated facts and verifying

---

[*]Work done during an internship at Apple.

whether each of these facts is fully supported by the input context, and could benefit significantly from a learned reasoning process.

Some existing hallucination detection works (Luo et al., 2023; Eliav et al., 2025) explored Chain-of-Thought (CoT) prompting, and showed that simple CoT can lead to considerable improvements in binary hallucination detection performance providing motivating evidence to explore reasoning for hallucination detection. However, these works do not focus on the fine-grained hallucination span detection problem and they do not explore training a reasoning model for hallucination detection. In this work, we focus on concretely answering the following two research questions: (i) Is learned reasoning process helpful for hallucination span detection? How to learn an effective reasoning process for this task? (ii) Is it necessary to learn a reasoning process specifically for hallucination span detection or do existing general-domain reasoning models suffice for this specific task?

To answer the first question, we train a CoT reasoning-based hallucination span detection model using Reinforcement Learning (RL). Specifically, we train the model on a dataset labeled with hallucination spans using Group Relative Policy Optimization (GRPO; Shao et al. (2024)) with a reward function based on the target span-F1 metric. To the best of our knowledge, this is the first work training a reasoning-based hallucination span detection model using RL. The resulting model significantly outperforms a non-reasoning model trained for span detection using Supervised Finetuning (SFT) on the same training dataset. This clearly shows that the reasoning process learned using RL is highly beneficial for detecting hallucination spans.

While the reward based on span-F1 score is effective, we notice that its asymmetric nature over-incentivizes non-hallucination predictions due to the normalization used in GRPO advantage calculation. To address this issue, we propose a modified version of GRPO, which we refer to as class-aware policy optimization, by introducing a scaling factor for the advantages computed for non-hallucination samples. By using a value smaller than one for this scaling factor, we are able to achieve a better balance between hallucination and non-hallucination classes leading to an overall higher span-F1 score.

To answer the second question, we evaluate several recent reasoning models that have been trained with data from various domains such as mathematics, coding, tool-calling, etc. Our evaluation results show that, despite being much larger in size, state-of-the-art reasoning models perform significantly worse than a 7B reasoning model trained specifically for hallucination span detection.

**Major contributions:** (i) We train a hallucination span detection model using reinforcement learning with span-level reward, and show that the resulting reasoning process improves the hallucination span detection performance by a significant margin when compared to a non-reasoning model trained with the same dataset. (ii) We show that existing reasoning models perform significantly worse when compared to a reasoning model specifically trained for hallucination span detection using RL with span-F1 reward. (iii) We identify an issue with span-F1 reward that leads to over-emphasis on non-hallucination predictions in the context of GRPO, and propose class-aware policy optimization to address this issue.

## 2 HALLUCINATION SPAN DETECTION

### 2.1 TASK

This paper focuses on the task of hallucination span detection in the context of Conditional Natural Language Generation (CNLG) tasks such as summarization and long-form question answering. Given the input context $c$ and the generated response $y = (y_1, y_2...y_T)$ consisting of $T$ characters, the goal is to identify all the hallucinated spans, which are text segments in $y$ that are not supported by $c$. Each hallucinated span $s$ is represented using its start and end positions in $y$.

### 2.2 MODEL

Existing works on hallucination span detection train either a decoder-based generative model that directly outputs hallucinated content as a list of text segments (Wu et al., 2023) or an encoder-based discriminative model that performs token-level binary classification (Ogasa & Arase, 2025). While generative models are a natural fit for exploring CoT reasoning, it is unclear how reasoning can

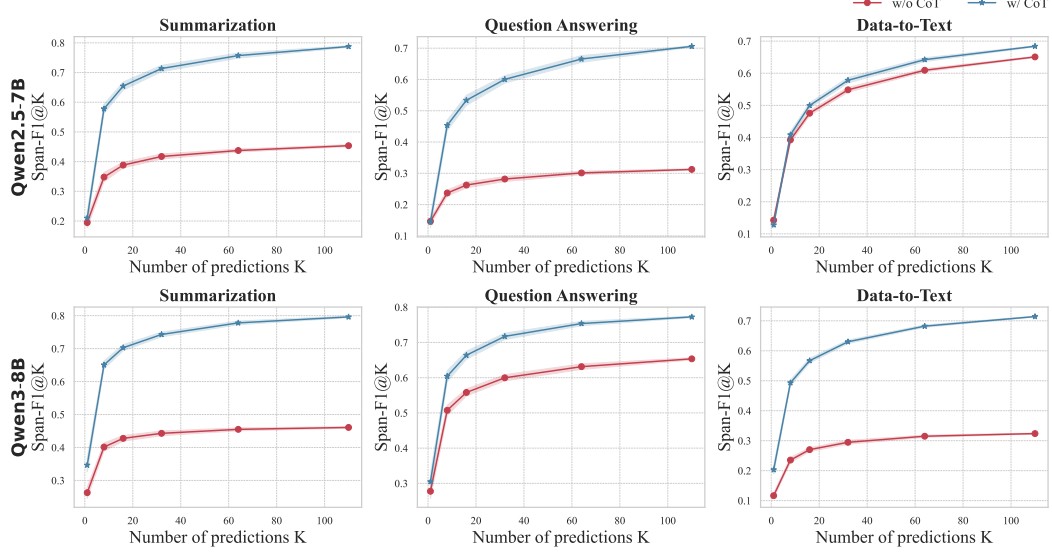

Figure 1: **Span-F1@K for different number of predictions** $K$**.** Using CoT reasoning provides significant boost as $K$ increases clearly demonstrating the potential of CoT reasoning.

be incorporated into token-level binary classifiers. Hence, in this work, we follow the generative modeling approach of Wu et al. (2023) and train an LLM to directly output a list of hallucinated text segments. For each predicted text segment, we get the corresponding span start and end index in $y$ by searching for matching content.

## 2.3 EVALUATION METRIC

For comparing model predictions with groundtruth, we use the dataset-level span-F1 metric defined in Wu et al. (2023). Given the groundtruth spans $S = \{s_m = [i_m, j_m]\}_{m=1}^{M}$ and the predicted spans $\hat{S} = \{s_n = [i_n, j_n]\}_{n=1}^{N}$, the span-F1 metric is computed using

$$\text{F1} = \frac{2 \cdot \text{Precision} \cdot \text{Recall}}{\text{Precision} + \text{Recall}}, \quad \text{Precision} = \frac{|\mathcal{P} \cap \mathcal{G}|}{|\mathcal{P}|}, \quad \text{Recall} = \frac{|\mathcal{P} \cap \mathcal{G}|}{|\mathcal{G}|}, \quad (1)$$

where $\mathcal{G} = \bigcup_{m=1}^{M} s_m$ and $\mathcal{P} = \bigcup_{n=1}^{N} s_n$. Here, $\cup$ denotes set union, $\cap$ denotes set intersection, $|.|$ denotes set cardinality, and $[i, j]$ denotes the set of integers from $i$ to $j$.

## 3 RL4HS: REINFORCEMENT LEARNING FOR HALLUCINATION SPAN DETECTION

### 3.1 MOTIVATING RL WITH DIVERSE CoT REASONING

A central question in this study is whether explicit reasoning is beneficial for identifying hallucination spans. As a preliminary experiment, we evaluated `Qwen2.5-7B` [1](Team, 2024) and `Qwen3-8B` [2] (Yang et al., 2025) models with and without CoT reasoning on data from three CNLG tasks, namely summarization, question answering and data-to-text using the RAGTruth dataset (Wu et al., 2023). In CoT reasoning mode, the model is encouraged to first reason about the factual consistency between the input context and the generated output, and then predict hallucinated spans. In the non-reasoning mode, the prompt given to the model instructs it to directly prediction hallucination spans without generating any intermediate content. For each input, the model is run $K$ times

---

[1]We use the instruct version.

[2]We use the reasoning mode and non-reasoning mode with non-COT prompt as elaborated in Qwen3.

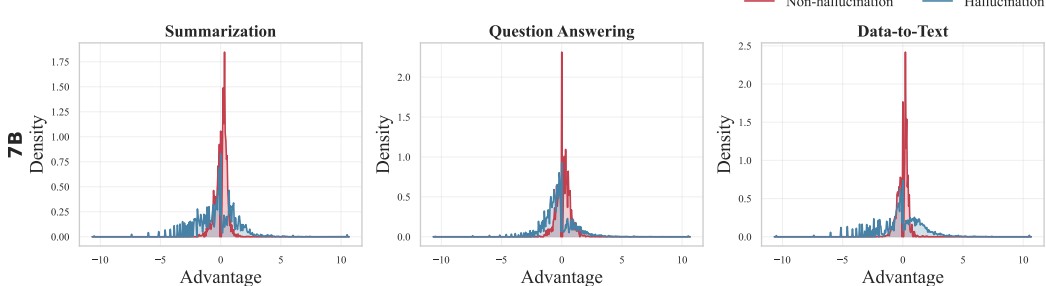

Figure 2: **Advantage distribution by model predictions.** Advantage distributions across tasks on Qwen2.5-7B-Instruct pretrained model. Non-hallucination predictions (red) receive higher advantages than hallucination predictions (blue), revealing a class imbalance issue.

and the best prediction is selected based on span-F1. We repeat this experiment for different values of $K$ and show the corresponding Span-F1@K results in Figure 1.

At $K = 1$, CoT reasoning provides no gains for `Qwen2.5-7B` and limited gains for `Qwen3-8B`. However, as $K$ increases, the gap in terms of Span-F1@K increases significantly demonstrating the potential of CoT reasoning to generate at least one accurate prediction when sampled multiple times. These results provide clear motivation to use reinforcement learning for bringing the reasoning capacity of LLMs related to hallucination span detection to the forefront.

We also conducted this experiment with `Qwen2.5-14B` and `Qwen3-14B` models and observed a similar behavior. See Appendix A.6 for details.

## 3.2 RL WITH GRPO

As our reinforcement learning framework, we employ Group Relative Policy Optimization (GRPO) Shao et al. (2024). Unlike Proximal Policy Optimization (PPO) Schulman et al. (2017), GRPO eliminates the explicit value function and instead computes baselines from relative group scores. The learning objective is defined as:

$$\mathcal{L}_{\text{GRPO}}(\theta) = \mathbb{E}_{\tau \sim \pi_\theta} \left[ \min \left( r_\theta(\tau) A(\tau), \, \text{clip}(r_\theta(\tau), 1 - \epsilon, 1 + \epsilon) \, A(\tau) \right) \right], \quad (2)$$

where $\tau$ denotes a trajectory sampled from the current policy $\pi_\theta$, and $r_\theta(\tau) = \frac{\pi_\theta(\tau)}{\pi_{\text{old}}(\tau)}$ is the probability ratio between the updated and reference policies at each step. Instead of relying on a critic network as in PPO, GRPO defines the advantage purely from group-based rewards $\{R_i\}_{i \in G(\tau)}$:

$$A(\tau) = \frac{R_\tau - \text{mean}\big(\{R_i\}_{i \in G(\tau)}\big)}{\text{std}\big(\{R_i\}_{i \in G(\tau)}\big)}. \quad (3)$$

In this formulation, the baseline is determined by the average performance of the group, normalized by its standard deviation, making GRPO particularly suited for scenarios where relative ranking within a group is more informative than absolute value estimates.

### 3.2.1 VERIFIABLE SPAN-F1 REWARD

To apply GRPO for hallucination span detection, we directly use the target span-F1 metric to define the reward. Let $\hat{S}$ be the predicted hallucination spans and $S$ be the ground-truth spans. Then, the reward is defined as

$$r_{\text{span}} = \begin{cases} 1, & \text{if } \hat{S} = \varnothing \text{ and } S = \varnothing, \\ \text{span-F1}(\hat{S}, S), & \text{otherwise.} \end{cases}$$

This formulation naturally handles both hallucination and non-hallucination cases. If no hallucinations exist and none are predicted, the model receives maximum reward ($r_{\text{span}} = 1$). In other cases, the reward reflects the quality of overlap between predicted and groundtruth spans.

## 3.3 REWARD IMBALANCE ACROSS CLASSES

Although GRPO normalizes advantages within groups, we find that the prediction type strongly biases the advantage values. As shown in Figure 2, predictions of non-hallucination consistently receive higher advantages than predictions of hallucination. Figure 3 shows the average advantage values by prediction type confirming that predicting non-hallucination is systematically rewarded more, independent of correctness.

This stems from an inherent asymmetry in the reward function $r_{span}$. In the non-hallucination class, a model only needs to predict an empty span list to obtain a high reward. In the hallucination class, the model must precisely localize and output the correct spans. This is a harder target, and small errors cause steep drops in the F1-based reward. As a result, GRPO tends to overincentivize non-hallucination predictions, leading to biased behaviors with high precision but suppressed recall.

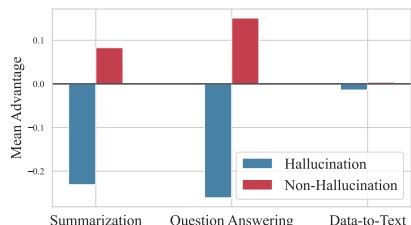

Figure 3: **Expected values of advantage given to Qwen2.5-7B-Instruct pretrained model predictions based on the prediction type.** Values are shown separately for the three task-based splits of the RAGTruth dataset.

## 3.4 CLASS-AWARE POLICY OPTIMIZATION

It may seem like a natural fix to the reward asymmetry issue is to use a smaller reward value for the case $\hat{S} = S = \varnothing$. However, the standardization step used in GRPO will eliminate the effect of such scaling. Hence, to address this imbalance issue, we introduce Class-Aware Policy Optimization (CAPO), which uses a scaling factor $\alpha$ to scale the advantage values computed for samples that belong to the non-hallucination class.

$$\hat{A}(\tau)^{(nh)} = \alpha \cdot \frac{R_\tau - \text{mean}\big(\{R_i\}_{i \in G(\tau)}\big)}{\text{std}\big(\{R_i\}_{i \in G(\tau)}\big)}.$$

This formulation balances the contributions of both classes, mitigating reward sparsity in non-hallucination examples and preventing dominance by non-hallucination examples. We use $\alpha = 0.5$ in our experiments. This value has been chosen based on the performance of trained model on a validation set.

## 4 EXPERIMENTAL SETUP

We design our experiments to answer the following research questions, which structure the results and discussion (Section 5): **Q1:** What is the effectiveness of RL4HS?; **Q2:** Does CAPO alleviate reward hacking and achieve better precision–recall balance?; **Q3:** Is in-Domain reasoning necessary for hallucination span detection?; **Q4:** Can simply scaling rewards solve reward hacking?; **Q5:** What does RL4HS learn?

**Dataset.** We conduct experiments on the **RAGTruth** benchmark Wu et al. (2023) as the statistics described in Table 6, which provides hallucination span annotations across three generation tasks: *Summarization*, *Question Answering (QA)*, and *Data-to-Text*. Each task contains paired source documents, model-generated responses, and human-labeled hallucination spans. This makes RAGTruth one of the few datasets suitable for training and evaluating hallucination detection at the span level rather than only binary classification.

**Models.** Our experiments primarily use the `Qwen2.5-7B-Instruct` and `Qwen2.5-14B-Instruct` models as base LLMs. For comparison, we additionally evaluate: **Pretrained reasoning models**: `Qwen3-8B`, `Qwen3-14B`, and `QwQ-32B`. **Proprietary reasoning models**: `GPT-5`, `o3`, `GPT-4o-mini` and `GPT-5-mini`. We use the default decoding strategy elaborated in the pre-trained models and use top-p = 0.95 (Holtzman et al., 2020), top-k = 20 (Holtzman et al., 2020), temperature = 0.6 for fine-tuned model generation.

**Baselines.** We compare RL4HS against the following approaches:

- **Supervised Fine-Tuning (SFT)** (Wu et al., 2023): trained with cross-entropy on hallucination span annotations.
- **RL4HS-GRPO**: our RL4HS approach but trained with GRPO instead of CAPO.
- **Multi-View Attention** (Ogasa & Arase, 2025): token-level detector using features aggregated from multiple attention heads and attention diversity views; evaluated on attention distributions across summarization and data-to-text tasks.

## 5 RESULTS & DISCUSSION

Table 1: **Span-level hallucination detection results on RAGTruth.** We report F1, precision, and recall across summarization, question answering, and data-to-text. Best scores are in bold. $^{\dagger}$ means the results taken from Ogasa & Arase (2025).

| Model | Summarization | | | Question Answering | | | Data-to-Text | | | Avg. | | |
|---|---|---|---|---|---|---|---|---|---|---|---|---|
| | F1 | Precision | Recall | F1 | Precision | Recall | F1 | Precision | Recall | F1 | Precision | Recall |
| *Proprietary models* | | | | | | | | | | | | |
| GPT-4o-mini w/ CoT | 38.4 | 43.4 | 34.4 | 27.3 | 33.7 | 23.0 | 33.7 | 34.2 | 33.2 | 33.1 | 37.1 | 30.2 |
| GPT-5-mini w/ CoT | 43.9 | 33.0 | 65.5 | 47.2 | 37.9 | 62.7 | 42.5 | 29.8 | 74.7 | 44.5 | 33.6 | 67.6 |
| GPT-5 w/ CoT | 36.5 | 24.9 | 68.4 | 44.4 | 32.1 | 71.8 | 45.7 | 33.2 | 73.5 | 42.2 | 30.0 | 71.2 |
| o3 w/ CoT | 48.5 | 40.7 | 60.1 | 49.9 | 43.4 | 58.9 | 55.2 | 45.6 | 70.0 | 51.2 | 43.2 | 63.0 |
| *Non-Reasoning models* | | | | | | | | | | | | |
| Qwen2.5-7B-Instruct w/o CoT | 19.3 | 28.9 | 14.5 | 14.7 | 19.2 | 11.9 | 14.0 | 22.3 | 10.2 | 16.0 | 23.5 | 12.2 |
| Qwen2.5-7B-Instruct w/ CoT | 21.0 | 27.4 | 17.1 | 14.5 | 18.8 | 11.7 | 13.0 | 32.5 | 8.2 | 16.2 | 26.2 | 12.3 |
| Qwen2.5-14B-Instruct w/o CoT | 31.5 | 28.0 | 36.2 | 27.8 | 50.7 | 55.8 | 29.0 | 22.8 | 39.8 | 29.4 | 33.8 | 43.9 |
| Qwen2.5-14B-Instruct w/ CoT | 32.9 | 44.4 | 26.1 | 22.6 | 29.6 | 31.6 | 26.3 | 45.0 | 18.6 | 27.3 | 39.7 | 25.4 |
| *Reasoning models* | | | | | | | | | | | | |
| QwQ-32B | 19.4 | 50.6 | 12.0 | 12.9 | 48.5 | 7.5 | 13.5 | 60.7 | 7.6 | 15.3 | 53.3 | 9.0 |
| Qwen3-8B | 34.7 | 42.2 | 29.5 | 30.5 | 32.0 | 29.1 | 20.3 | 45.2 | 13.1 | 28.5 | 39.8 | 23.9 |
| Qwen3-14B | 35.8 | 36.9 | 34.9 | 30.6 | 30.7 | 30.6 | 34.8 | 40.9 | 30.4 | 33.7 | 36.2 | 32.0 |
| *Finetuned models* | | | | | | | | | | | | |
| SFT-7B | 44.1 | 52.2 | 38.2 | 51.3 | 51.3 | 51.4 | 54.8 | 58.8 | 51.5 | 50.1 | 54.1 | 47.0 |
| SFT-14B | 52.7 | 57.6 | 48.7 | 53.9 | 53.1 | 54.8 | 59.6 | 61.6 | 57.8 | 55.4 | 57.4 | 53.8 |
| Multi-View Attention-7B$^{\dagger}$ | 41.5 | 49.6 | 35.7 | 50.6 | 38.5 | 73.7 | 55.2 | 53.5 | 57.1 | 49.1 | 47.2 | 55.5 |
| *Ours: RL4HS* | | | | | | | | | | | | |
| RL4HS-GRPO-7B | 51.2 | 68.7 | 40.9 | 55.0 | 59.6 | 52.1 | 56.3 | 66.5 | 48.8 | 54.2 | 64.9 | 47.3 |
| RL4HS-7B | 50.9 | 64.4 | 42.3 | **56.4** | 57.1 | 56.5 | 60.4 | 67.1 | 54.9 | 55.9 | 62.9 | 51.2 |
| RL4HS-14B | **57.6** | 64.2 | 52.3 | 54.8 | 52.5 | 57.3 | **62.6** | 67.2 | 58.7 | **58.3** | 61.3 | 56.1 |

### 5.1 **Q1:** WHAT IS THE EFFECTIVENESS OF RL4HS?

Table 1 reports span-level hallucination detection results on RAGTruth across summarization, question answering, and data-to-text. We compare pretrained prompting baselines with models fine-tuned under our RL4HS framework.

**Pretrained instruction-tuned models.** Qwen2.5-7B/14B-Instruct, with or without CoT, perform poorly (F1 below 30), indicating that prompting alone is insufficient for accurate span localization.

**Pretrained reasoning models.** Models designed for reasoning (QwQ-32B, Qwen3-8B, Qwen3-14B) transfer some reasoning ability to hallucination detection. For example, Qwen3-14B improves summarization F1 to 35.8 compared to 32.9 for Qwen2.5-14B-Instruct. However, these models still trail fine-tuned approaches, showing that general reasoning ability alone is insufficient for span-level detection.

**Finetuned baselines.** Supervised fine-tuning (SFT) provides strong gains, reaching 55.4 F1 at 14B scale. Multi-View Attention (Ogasa & Arase, 2025) further pushes the 7B model to 49.1 F1, though still behind larger SFT models.

**RL4HS** RL4HS consistently outperforms all baselines, including proprietary GPT-4o/5-mini, GPT-5, and o3. RL4HS-7B outperforms SFT on all three tasks (avg. 55.9 v.s 50.1). At 14B, RL4HS-14B achieves 57.6 on summarization, 54.8 on QA, and 62.6 on Data-to-Text, surpassing Qwen3 and the strongest GPT-5 and o3 models. This establishes RL4HS demonstrating that reinforcement learning with span-level rewards effectively aligns reasoning with hallucination detection.

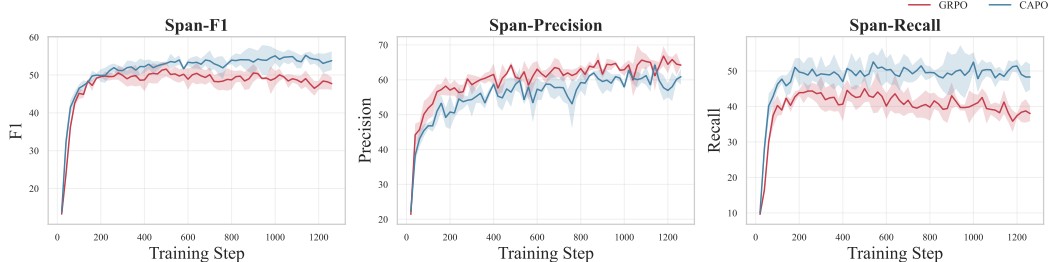

Figure 4: **Training dynamics of GRPO (red) and CAPO (blue) on Qwen2.5-7B-Instruct model.** While GRPO exhibits high precision but declining recall due to reward hacking, CAPO stabilizes recall without sacrificing precision, yielding consistently higher span F1. Shaded regions denote standard deviations across runs.

## 5.2  Q2: DOES CAPO ALLEVIATE REWARD HACKING AND ACHIEVE BETTER PRECISION−RECALL BALANCE?

A key limitation we observed with GRPO is that models often exploit the reward design by defaulting to predicting no hallucination spans, which yields high precision but severely hurts recall. This behavior reflects a form of *reward hacking*, where the model learns shortcuts that maximize rewards without genuinely improving hallucination detection. As shown in our advantage distribution analysis (Figure 2), predictions of non-hallucination systematically receive higher advantages, biasing the policy toward conservative behavior.

Figure 4 compares training dynamics of GRPO and our proposed CAPO across span F1, precision, and recall. We make two observations: (1) **GRPO favors precision over recall.** As training progresses, GRPO maintains relatively high precision but recall gradually drops, showing the model's tendency to avoid making positive span predictions.; (2) **CAPO balances precision and recall.** By re-weighting class-specific advantages, CAPO stabilizes recall while preserving strong precision, resulting in a clear improvement in span F1 throughout training.

These results confirm that CAPO directly addresses the imbalance highlighted in our advantage distribution analysis. By correcting for class-dependent reward sparsity, CAPO mitigates reward hacking and achieves a better precision–recall trade-off, consistently yielding higher span F1 compared to vanilla GRPO.

## 5.3  Q3: IS IN-DOMAIN REASONING NECESSARY FOR HALLUCINATION SPAN DETECTION?

To assess whether hallucination span detection requires in-domain reasoning rather than generic reasoning ability, we conduct leave-one-out training with RL4HS (RL4HS-OOD-7B), holding out one task at a time and evaluating on the unseen task. Figure 5 shows results compared against reasoning-focused models (QwQ, Qwen3) and large-scale GPT-series baselines.

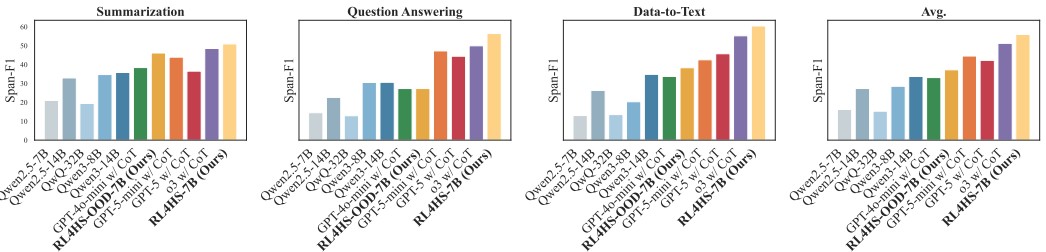

Figure 5: **Out-of-domain evaluation on RAGTruth.** Span-F1 scores on Ragtruth dataset. Our RL4HS-OOD-7B model performs competitively with larger reasoning models, showing the benefit of span-level reward fine-tuning. We use Instruct models for Qwen2.5 models.

General-purpose reasoning models such as Qwen3 and QwQ transfer some reasoning ability but their Span-F1 scores often remain below 40, showing that generic reasoning is insufficient for fine-grained hallucination detection. RL4HS-OOD-7B, in contrast, achieves consistently stronger results across all held-out tasks, approaching the in-domain "task" topline RL4HS-7B. Moreover, despite being much smaller, RL4HS-OOD-7B performs better than GPT-4-mini and remains competitive with GPT-5-mini, and even GPT-5, underscoring the efficiency of span-level reward fine-tuning. These findings highlight that in-domain "reasoning" learned with span-level rewards is essential for robust hallucination detection.

### 5.4 Q4: CAN SIMPLY SCALING REWARDS SOLVE REWARD HACKING?

One concern with GRPO is that its standardization of group rewards diminishes the intrinsic difficulty difference between hallucination and non-hallucination cases, often biasing the model toward predicting non-hallucination. To address this, we explored a variant of Dr.GRPO (Liu et al., 2025), which removes standardization and instead scales the reward for successfully predict non-hallucination by a factor $\gamma$. Table 2 reports results under different $\gamma$ values. While Dr.GRPO influences the precision–recall tradeoff (e.g., higher $\gamma$ increases recall at the cost of precision), overall performance is inferior to standard GRPO and RL4HS. This suggests that the normalization step in GRPO is crucial, and simple reward rescaling cannot effectively address reward hacking in our task.

Table 2: **Comparison of GRPO, CAPO, and Dr.GRPO variants with RL4HS.** CAPO improves F1 by addressing reward imbalance, while Dr.GRPO with different $\gamma$ values shows varying precision–recall trade-offs but does not surpass CAPO.

| Method | | Avg. | |
|---|---|---|---|
| | F1 | Precision | Recall |
| GRPO | 54.2 | 64.9 | 47.3 |
| CAPO | **55.9** | 62.9 | 51.2 |
| Dr.GRPO w/ $\gamma$=0.1 | 52.5 | 53.6 | 52.3 |
| Dr.GRPO w/ $\gamma$=0.5 | 54.7 | 62.2 | 49.4 |
| Dr.GRPO w/ $\gamma$=1.0 | 53.1 | 64.1 | 45.8 |

### 5.5 Q5: WHAT DOES RL4HS LEARN? A CASE STUDY

To better understand the reasoning behaviors learned by RL4HS, we examine qualitative outputs on the RAGTruth dataset (Table 3). The example highlights a discrepancy regarding whether the restaurant provides catering services. **Pretrained model.** Before training, the pretrained model fails to identify the inconsistency. Although it checks structured business hours and customer reviews, it overlooks the fact that the structured data contains no attribute related to catering services. As a result, the model produces no hallucination spans. **RL4HS.** In contrast, RL4HS correctly flags the catering services claim as a hallucination. Its reasoning process closely mirrors the human-designed heuristic pipeline:

- Step 1: Identify explicit claims in the article (e.g., "provides catering services").
- Step 2: Cross-check these claims against structured business data (which does not list catering services as an attribute).
- Step 3: Conclude that the claim is inconsistent and mark it as hallucinated.

This case demonstrates that RL4HS goes beyond surface-level reasoning traces. Instead of producing generic or irrelevant explanations, the model performs systematic consistency checks that align with heuristic rules used in prior hallucination detection pipelines. This suggests that the reasoning behavior learned under span-level rewards is genuine, faithful, and semantically grounded.

## 6 RELATED WORKS

**Hallucination Detection.** Hallucination detection research has evolved from binary classification to fine-grained span detection. Early work focused on binary judgments—whether text contains hallucinations (Manakul et al., 2023; Luo et al., 2023; Tang et al., 2024). However these approach failed to localize where the hallucination. Yang et al. (2024b); Scirè et al. (2024) proposed a cascade pipeline that leverage atomic-fact generation, natural language inference to detection hallucination. But the pipeline is hard to optimize. Recent methods target span-level detection. introduced

Table 3: **Case study comparing pretrained and RL4HS models on detecting hallucinations.**

---

*Review data*

'name': 'Benchmark Eatery', 'address': '1201 State St', 'city': 'Santa Barbara', 'state': 'CA', 'categories': 'American (Traditional), American (New), Breakfast & Brunch, Restaurants, Seafood, Vegetarian, Nightlife, Event Planning & Services, Bars, Venues & Event Spaces', 'hours': 'Monday': '0:0 0:0', 'Tuesday': '11:30-20:0', 'Wednesday': '11:30-20:0', 'Thursday': '11:30-20:0', 'Friday': '11:30-16:0', 'Saturday': '11:30-16:0', 'Sunday': '11:30-20:0', 'attributes': 'BusinessParking': 'garage': True, 'street': True, 'validated': False, 'lot': True, 'valet': False, 'RestaurantsReservations': False, 'OutdoorSeating': True, 'WiFi': 'free', 'RestaurantsTakeOut': True, 'RestaurantsGoodForGroups': True, 'Music': False, 'Ambience': 'touristy': False, 'hipster': False, 'romantic': False, 'divey': False, 'intimate': False, 'trendy': False, 'upscale': False, 'classy': True, 'casual': True, 'business stars': 4.0, 'review info': ['review stars': 4.0, 'review date': '2022-01-02 21:02:49', 'review text': 'Nice little place in downtown Santa Barbara where we stopped for late lunch/early dinner on our way back home to San Diego. We loved our flatbreads. I had been craving clam chowder and ordered it here. It was a little thick for my liking, but tasted good. Nice, friendly upscale casual place on State Street.', 'review stars': 1.0, 'review date': '2021-12-20 22:29:32', 'review text': "I had high hopes for this place, but it fell flat big time. I ordered the BLAT with grilled chicken and a side salad. I specifically asked for no tomato and they put tomato on both the sandwich and the salad. Not a huge deal, just annoying to pick it all off. And if you have an allergy, I would not trust this place to be mindful of that. The grilled chicken was not seasoned at all and the food overall was just extremely bland. My sandwich was stacked so high that the avocado all fell out as I was eating it. The side salad is lettuce, cucumber, red onion, and tomato. My bf got a burger with onion rings and he did not enjoy either part of the meal. The onion rings are beer battered and basically all batter. The burger was also unseasoned and not enjoyable. The service is ok. The staff is nice enough but my bf did have to get up and ask for his soda to be refilled and we were never offered a refill on our waters. We both left the restaurant with a stomach ache, I personally felt ill. I had stomach pains for a few hours afterwards and I can only assume it's because the food is not fresh because my meal didn't have any allergens that would've given me that reaction. Lastly, the cleaning spray they used on the tables just smelled funky. Idk how to explain it but we actually moved to an outside table because the inside one smelled so musty and mildewy.patio is nice, but not worth coming back for. Additionally, the menu is too scattered and overwhelming. I think they should stick to one type of food and work on increasing the quality rather than having too many different menu options that aren't cohesive.", 'review stars': 5.0, 'review date': '2021-11-29 20:45:43', 'review text': 'We have been here a few times during trial over the past couple weeks and it is awesome. I keep ordering the burrito but everyone seems to enjoy their food. The one piece of feedback would be the Spanish soup seems to change - from not too spicy to spicy.']

---

*Machine-generated article given review data*

Benchmark Eatery is an American-style restaurant located in downtown Santa
Barbara, California. The restaurant offers a variety of cuisines, including traditional American, seafood, vegetarian, and event planning services. They also provide outdoor seating, free Wi-Fi, takeout options, and catering services . According to customer reviews, the restaurant has a four-star rating, with mixed opinions about the quality of the food and service. Some customers have praised the restaurant's flatbreads and clam chowder, while others have criticized the restaurant for poor flavor, blandness, and inconsistent ingredients. Additionally, some customers have expressed concerns about the restaurant's handling of allergies and the cleanliness of the establishment. Overall, Benchmark Eatery appears to be a popular choice for dining in downtown Santa Barbara, with a range of menu options and a cozy atmosphere.

---

*Before RL4HS-7B (Pretrained model)*

## Step 1: Analyze the structured data to understand the restaurant's categories and services .
The restaurant is categorized as offering American (Traditional) and American (New) cuisine, as well as Breakfast & Brunch. It also provides seafood, vegetarian options, event planning services, outdoor seating, free Wi-Fi, takeout options, and catering services. ## Step 2: Compare the information in the article with the structured data. The article correctly mentions that Benchmark Eatery is an American-style restaurant located in downtown Santa Barbara. It also correctly states that they offer a variety of cuisines including traditional American and seafood. The article accurately notes that they provide outdoor seating and free Wi-Fi. ## Step 3: Check for any discrepancies between the review information provided in structured data and mentioned in the article. The review information includes both positive feedback about flatbreads and clam chowder from some customers, as well as negative feedback about poor flavor, blandness from others. The article accurately captures these mixed opinions. ## Step 4: Verify if there are any inconsistencies regarding food handling practices or cleanliness. The structured data does not explicitly mention concerns about handling allergies or cleanliness issues; however, one customer review did express concern about inconsistent ingredients used in their meal which could imply some issues with food quality or preparation consistency. The article mentions concerns about handling allergies but does not specifically state cleanliness issues based on this data alone. ## Step 5: Confirm overall rating based on customer reviews. The business has a four-star rating according to customer reviews mentioned in both sources. Based on this analysis: {"hallucination list": []}

---

*RL4HS-7B*

## Step 1: Check for Catering Services The article states "They also provide outdoor seating, free Wi-Fi, takeout options, and catering services." However, per data, catering services seems inconsistent. It seems to say "catering services" which seems inconsistent. It appears to be an error; suspecting potential candidates of typo or misinterpretation. ## Step 2: Evaluate Food Quality and Services. The text says that according to customer's opinion's the restaurant has a four-star rating with mixed food quality and service as discussed. ## Step 3: Validate Food Quality's. It states that "Some customers have praised the restaurant's flatbreads and clam chowder, while others have criticized the restaurant for poor flavor, blandness, and inconsistent ingredients." This part is confirmed by our data. ## Step 4: Evaluate Customer's Concerns. It says customers expressed concerns about food quality again as well as cleanliness of establishment; this part seems valid per our data. ## Step 5: Check for Food Variety. It describes variety of cuisines like traditional American (both traditional & new), breakfast & brunch, seafood, vegetarian food etc. again as said in text. ## Step 6: Evaluate Overall Assessment of Restaurant's Attraction. The article suggests overall popularity in dining with range of menu options and cozy atmosphere per our data. Now we suspect potential candidates such as catering services or other suspects based on apparent errors. If it says catering services it seems suspicious. The candidate suspects are often candidates of typographical errors or misinterpretation: ### Compiled Results: {"hallucination list": ["catering services"]}

---

RAGTruth (Wu et al., 2023) with human-annotated spans across three generation tasks. Ogasa & Arase (2025) aggregated multi-head attention features for token-level detection. However, these attention-based methods lack explicit reasoning mechanisms.

**Reasoning Enhancement in NLP.** Group Relative Policy Optimization (GRPO), originally developed to improve mathematical reasoning by comparing groups of outputs rather than relying on a separate value model. GRPO has since been extended and adapted to a variety tasks such coding (Liu & Zhang, 2025; Chen et al., 2025), planning (Hao et al., 2023), tool-calling (Feng et al., 2025a; Shang et al., 2025). More recently, researchers has show that GRPO can also be applied to enhance reasoning in traditional NLP tasks such as NLI (Shao et al., 2024), intent classification (Feng et al., 2025b), and safety alignment Li et al. (2025). Showing the effectiveness of GRPO with LLM.

## 7 CONCLUSION

We introduced RL4HS, a reinforcement learning framework that uses span-level rewards to align LLM reasoning with hallucination detection. While CoT offers limited single-sample gains, RL4HS distills its multi-sample advantages into stronger predictions. With CAPO to address reward imbalance, RL4HS outperforms pretrained reasoning models and SFT on RAGTruth, and produces faithful, heuristic-like reasoning traces that improve both accuracy and robustness.

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

## A  APPENDIX

### A.1  USE OF LARGE LANGUAGE MODELS (LLMS)

In preparing this paper, we made limited use of a large language model (LLM) as a general-purpose writing assistant. Specifically, the LLM was employed to refine the clarity, grammar, and flow of the manuscript text. The LLM did not contribute to the research ideation, methodology, experimental design, analysis, or interpretation of results. All scientific content, claims, and conclusions are solely the responsibility of the authors.

## A.2  PROMPT

---

**COT for Summarization**

"Below is the original document:"
{reference}
"Below is a summary of the document:"
{response}
"Your task is to determine whether the summary contains hallucinations." "First, provide reasoning with the following format:"
## Step 1: < your first reasoning step >
## Step 2: < your next reasoning step >
...(add as many steps as needed) Then, compile the labeled hallucinated spans into a JSON dict, with a key hallucination list and its value is a list of hallucinated spans. If there are potential hallucinations, the output should be in the following JSON format: hallucination list: [hallucination span1, hallucination span2, ...]. Otherwise, leave the value as an empty list as follows: hallucination list: [].

---

**COT for Question Answering**

"Below is a question:"
{question}
"Below are the related passages:"
{reference}
"Below is an answer:"
{response}
"Your task is to determine whether the answer contains hallucinations." "First, provide reasoning with the following format:"
## Step 1: < your first reasoning step >
## Step 2: < your next reasoning step >
...(add as many steps as needed) Then, compile the labeled hallucinated spans into a JSON dict, with a key hallucination list and its value is a list of hallucinated spans. If there are potential hallucinations, the output should be in the following JSON format: hallucination list: [hallucination span1, hallucination span2, ...]. Otherwise, leave the value as an empty list as follows: hallucination list: [].

---

**COT for Data-to-text**

"Below is structured data in JSON format:"
{reference}
Below is an overview article written in accordance with the structured data:"
{response}
"Your task is to determine whether the article contains hallucinations." "First, provide reasoning with the following format:"
## Step 1: < your first reasoning step >
## Step 2: < your next reasoning step >
...(add as many steps as needed) Then, compile the labeled hallucinated spans into a JSON dict, with a key hallucination list and its value is a list of hallucinated spans. If there are potential hallucinations, the output should be in the following JSON format: hallucination list: [hallucination span1, hallucination span2, ...]. Otherwise, leave the value as an empty list as follows: hallucination list: [].

---

---

**w/o COT for Summarization**

"Below is the original document:"
{reference}
"Below is a summary of the document:"
{response}
"Your task is to determine whether the summary contains hallucinations."
Then, compile the labeled hallucinated spans into a JSON dict, with a key hallucination list and its value is a list of hallucinated spans. If there are potential hallucinations, the output should be in the following JSON format: hallucination list: [hallucination span1, hallucination span2, ...]. Otherwise, leave the value as an empty list as follows: hallucination list: [].

---

**w/o COT for Question Answering**

"Below is a question:"
{question}
"Below are the related passages:"
{reference}
"Below is an answer:"
{response}
"Your task is to determine whether the answer contains hallucinations."
Then, compile the labeled hallucinated spans into a JSON dict, with a key hallucination list and its value is a list of hallucinated spans. If there are potential hallucinations, the output should be in the following JSON format: hallucination list: [hallucination span1, hallucination span2, ...]. Otherwise, leave the value as an empty list as follows: hallucination list: [].

---

**w/o COT for Data-to-text**

"Below is structured data in JSON format:"
{reference}
Below is an overview article written in accordance with the structured data:"
{response}
"Your task is to determine whether the article contains hallucinations."
Then, compile the labeled hallucinated spans into a JSON dict, with a key hallucination list and its value is a list of hallucinated spans. If there are potential hallucinations, the output should be in the following JSON format: hallucination list: [hallucination span1, hallucination span2, ...]. Otherwise, leave the value as an empty list as follows: hallucination list: [].

---

A.3 TRAINING DETAILS

Table 4: **Training details for SFT and RL.**

| Method | Size | Learning Rate | Batch Size |
|--------|------|---------------|------------|
| SFT | 7B | 1e-6 | 64 |
|     | 14B | 1e-6 | 64 |
| RL | 7B | 1e-6 | 64 |
|    | 14B | 5e-7 | 64 |

We trained our models using 8 H100 GPUs. The learning rate and batch size configurations are provided in Table 4. For reinforcement learning training, we set the group size to 16 and used rollout generation with temperature = 1.0, top-p = 1.0, and top-k = -1. Following Yu et al. (2025),

we also adopted a clipping threshold of clip_high = 0.28. Due to the lack of the reasoning data, we fine-tuned instruct model with RL directly instead of doing SFT first.

For GPT-series models, we used top-p = 0.95 and temperature = 0.7 to generate response during inference. All the trained models were trained with 5 epochs and selected the checkpoints with the best performance on self-splitted validation set. In our training, we resolved the data class imbalance by upweighting hallucination class to have equal amount of data.

## A.4 STANDARD DEVIATION

Table 5: **Span-level hallucination detection results (STD) on RAGTruth.**

| Model | Summarization | | | Question Answering | | | Data2Text | | |
|---|---|---|---|---|---|---|---|---|---|
| | F1 | Precision | Recall | F1 | Precision | Recall | F1 | Precision | Recall |
| *Proprietary models* | | | | | | | | | |
| GPT-4o-mini w/ COT | 1.5 | 1.9 | 1.5 | 0.6 | 0.7 | 0.8 | 1.9 | 2.1 | 1.9 |
| GPT-5-mini w/ COT | 1.0 | 0.9 | 1.4 | 0.5 | 0.4 | 0.9 | 1.5 | 1.1 | **2.5** |
| GPT-5 w/ COT | 0.5 | 0.4 | 1.2 | 0.4 | 0.4 | 1.1 | 0.9 | 0.6 | 1.7 |
| O3 w/ COT | 0.8 | 0.8 | 1.4 | 0.5 | 0.5 | 0.8 | 2.1 | 1.5 | 3.3 |
| *Non-Reasoning models* | | | | | | | | | |
| Qwen2.5-7B-Instruct w/o COT | 1.7 | 2.5 | 1.4 | 1.1 | 1.3 | 0.9 | 1.1 | 1.3 | 0.9 |
| Qwen2.5-7B-Instruct w/ COT | 2.0 | 2.7 | 1.8 | 1.1 | 1.3 | 0.9 | 1.0 | 2.6 | 0.7 |
| Qwen2.5-14B-Instruct w/o COT | 1.3 | 1.3 | 1.5 | 0.7 | 0.7 | 0.9 | 0.6 | 0.6 | 0.7 |
| Qwen2.5-14B-Instruct w/ COT | 2.3 | 2.8 | 2.2 | 1.8 | 3.0 | 1.4 | 1.3 | 1.9 | 1.1 |
| *Reasoning models* | | | | | | | | | |
| QwQ-32B | 1.9 | 3.8 | 1.3 | 1.4 | 4.0 | 0.9 | 1.0 | 2.6 | 0.6 |
| Qwen3-8B | 2.1 | 2.4 | 2.2 | 2.2 | 1.8 | 2.6 | 1.4 | 2.7 | 1.0 |
| Qwen3-14B | 1.8 | 2.0 | 1.8 | 1.9 | 1.8 | 2.1 | 1.1 | 1.2 | 1.3 |
| *Finetuned models* | | | | | | | | | |
| SFT-7B | 1.4 | 2.8 | 0.6 | 0.3 | 1.6 | 1.2 | 0.4 | 2.4 | 2.4 |
| SFT-14B | 1.2 | 3.2 | 0.5 | 0.7 | 1.0 | 0.5 | 0.6 | 1.8 | 0.5 |
| Multi-View Attention-7B | N/A | N/A | N/A | N/A | N/A | N/A | N/A | N/A | N/A |
| *Ours: RL4HS* | | | | | | | | | |
| RL4HS-GRPO-7B | 1.0 | 2.2 | 1.3 | 2.4 | 7.5 | 4.5 | 0.5 | 0.5 | 0.4 |
| RL4HS-7B | 0.5 | 0.5 | 0.4 | 4.4 | 4.8 | 7.3 | 0.3 | 1.5 | 1.6 |
| RL4HS-14B | 1.1 | 1.0 | 1.9 | 1.1 | 1.0 | 1.9 | 0.7 | 1.7 | 2.4 |

## A.5 DATASET STATISTIC

Table 6: **Dataset statistics for RAGTruth.** Numbers indicate the number of hallucination examples, with the number of non hallucination examples shown in parentheses.

| | Summarization | Question Answering | Data-to-Text |
|---|---|---|---|
| Train | 1209 (2646) | 1277 (2732) | 3048 (1347) |
| Val | 271 (629) | 269 (614) | 624 (276) |
| Test | 204 (696) | 160 (715) | 579 (321) |

## A.6 SPAN-F1@K

More results on Span-F1@K for Qwen2.5 (Figure 6) and Qwen3 (Figure 7) models.

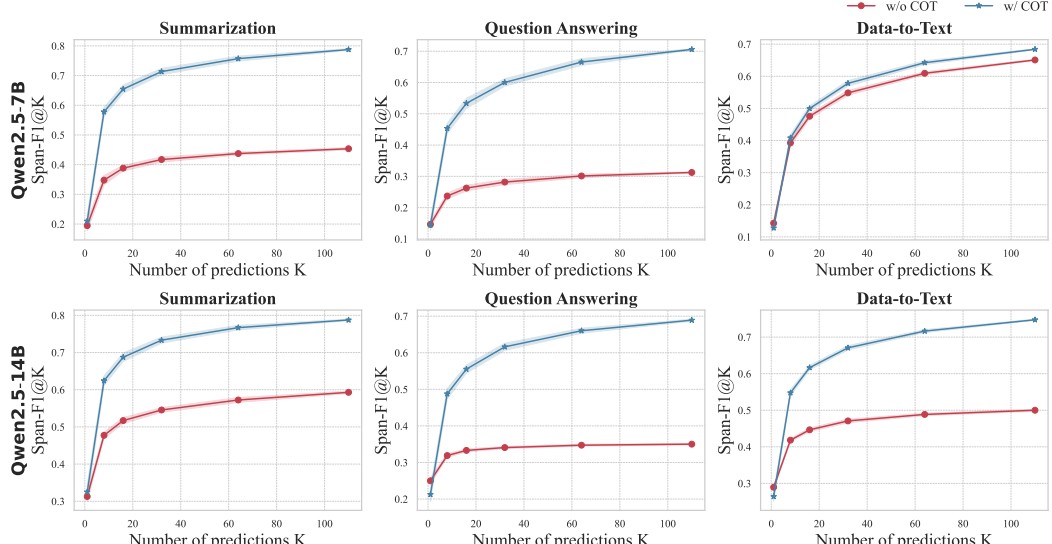

Figure 6: Hallucination span detection with and without CoT reasoning. Results are shown for summarization, question answering, and data-to-text tasks on the RAGTruth benchmark.

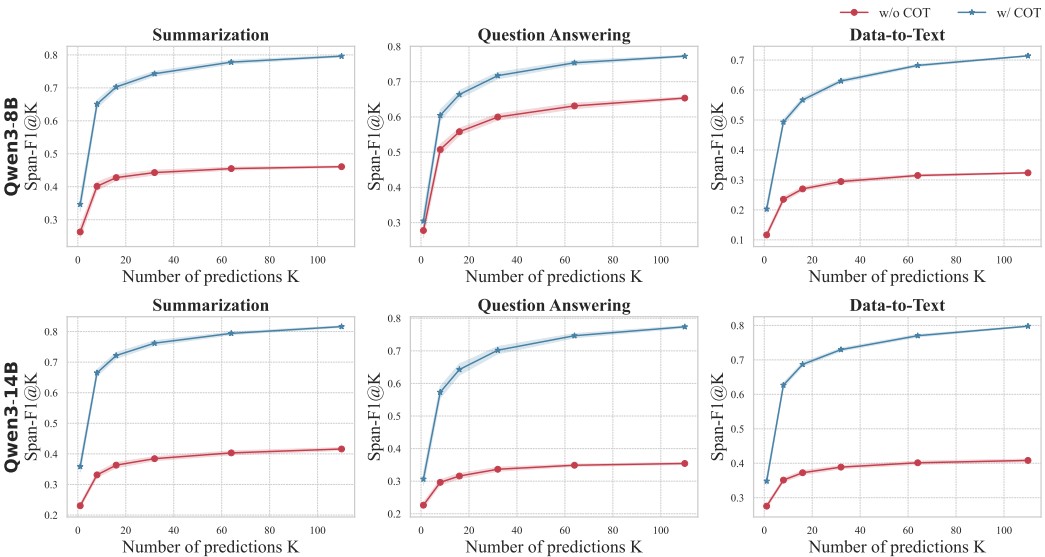

Figure 7: Hallucination span detection with and without CoT reasoning. Results are shown for summarization, question answering, and data-to-text tasks on the RAGTruth benchmark.

## A.7 MORE RL4HS REASONING EXAMPLES

We provide three additional examples of RL4HS reasoning paths in Table 7, 8 and 9. The third example illustrates a common mistake of RL4HS. Specifically, the RL4HS model becomes confused while converting between 24hr clock to 12hr clock formats.

### Table 7: **RL4HS reasoning example 1 (QA)**

| |
|---|
| *Question* |
| differences and similarities between red and white blood cells |
| *Retrieved passages* |
| **passage 1:** Shares & Saves. Save. Blood is made up of three parts: red cells, white cells and platelets. Red blood cells are responsible for the oxygen delivery from the lungs to the tissue in the body. White blood cells are a part of the immune system. These components circulate in the fluids in the blood vessels, which is pumped by the heart.hares & Saves. Save. Blood is made up of three parts: red cells, white cells and platelets. Red blood cells are responsible for the oxygen delivery from the lungs to the tissue in the body. White blood cells are a part of the immune system. |
| **passage 2:** 1. Both red blood cells and white blood cells are found in blood. 2. Both red blood cells and white blood cells can be carried by blood from one part to other part of our body . . . .3.The red blood cells and the White Blood Cells can be found in the circulatory system. 4.It both have cell membrane and cytoplasm.hite blood cells are part of the Immune System and help the body combat disease, viruses, etc. Red blood cells are part of the Circulatory System and carry things such as oxy . . . gen throughout the body. |
| **passage 3:** Best Answer: Similarities: - RBCs, WBCs are formed elements of blood. - RBCs, WBCs are formed from pluripotent stem cells in red bone marrow.est Answer: Similarities: - RBCs, WBCs are formed elements of blood. - RBCs, WBCs are formed from pluripotent stem cells in red bone marrow. |
| *Machine-generated answer given question and retrieved passages* |
| Based on the given passages, here are the differences and similarities between red and white blood cells: Differences: Function: Red blood cells are responsible for delivering oxygen from the lungs to the body's tissues, while white blood cells are part of the immune system and help combat diseases and infections. Structure: Red blood cells have a unique biconcave disk shape that allows them to maximize their surface area for gas exchange, whereas white blood cells have a irregular, amoeba-like shape. Similarities: Formation: Both red and white blood cells are formed from pluripotent stem cells in the red bone marrow. Presence in circulatory system: Both red and white blood cells are found in the circulatory system and can be carried by blood from one part of the body to another. Cellular components: Both red and white blood cells have a cell membrane and cytoplasm. Therefore, while red and white blood cells have some similarities in terms of their formation and presence in the circulatory system, they differ in their function and structure. |
| *RL4HS reasoning path and output* |
| ## Step 1: The statement 'Differences:' with '* Function: Red blood cells are responsible for delivering oxygen from the lungs to the body's tissues, while white blood cells are part of the immune system and help combat diseases and infections.' based from Passage 1 and Passage 2 seems accurate from provided content about their roles directly. ## Step 2: The statement '* Structure: Red blood cells have a unique biconcave disk shape that allows them to maximize their surface area for gas exchange, whereas white blood cells have a irregular, amoeba-like shape.' appears without strong basis from any of given related passages (Passage1, Passage2 or Passage3). None of these text snippets confirm details about structure or shapes directly. ## Step 3: The statement 'Similarities:' with content about formation based off passage3 seem accurate. Details provided from snippet directly. ## Step 4: The statement 'Presence in circulatory system:' along with 'Cellular components:' seem based off passage2 accurately. {{"hallucination list": ["* Structure: Red blood cells have a unique biconcave disk shape that allows them to maximize their surface area for gas exchange, whereas white blood cells have a irregular, amoeba-like shape."]}} |

### Table 8: **RL4HS reasoning example 2 (QA)**

| |
|---|
| *Question* |
| what is the proper way to fertilize your grass |
| *Retrieved passages* |
| **passage 1:** Fertilize at the right time of year. In order to your fertilizer to work best, you'll need to time it so that you fertilize at the beginning of the growing season. This is different for different types of grasses. Fertilize warm season grasses when the grass begins to turn green at the beginning of spring. |
| **passage 2:** Picking Out Fertilizer. 1 1. Know your grass. The type of grass you have will determine what type of fertilizer you need to use, and how often you need to fertilize. 2 2. Do a pH soil test. This measures the alkalinity or acidity of the soil, and helps to determine what pH your fertilizer should have. 3 3. Measure your lawn. |
| **passage 3:** Pour in the rest of the fertilizer. Apply the rest of fertilizer in a perpendicular pattern. Step on the turf and if the leaf blade bounces back up, your lawn is in good shape. If it lays flat, you've over-fertilized. |
| *Machine-generated answer given question and retrieved passages* |
| Based on the provided passages, the proper way to fertilize your grass is as follows: Time your fertilization at the beginning of the growing season, which varies for different types of grasses. Warm season grasses should be fertilized when the grass turns green in the early spring, while cool season grasses should be fertilized in the fall. (Passage 1) Choose a fertilizer that is appropriate for your specific type of grass and perform a pH soil test to determine the correct pH level for your fertilizer. (Passage 2) Apply the remaining fertilizer in a perpendicular pattern, and check if the leaf blade bounces back up after stepping on the turf to ensure that your lawn is in good shape. If the leaf blade lies flat, you may have over-fertilized. (Passage 3) Unable to answer based on given passages. The passages do not provide information on the amount of fertilizer to use, how often to fertilize, or any specific guidelines for organic or slow-release fertilizers. |
| *RL4HS reasoning path and output* |
| ## Step 1: The statement '1. Time your fertilization at the beginning of the growing season, which varies for different types of grasses. Warm season grasses should be fertilized when the grass turns green in the early spring, while cool season grasses should be fertilized in the fall. (Passage 1)' from answer seems to include about cool season grasses part about being fertilized in fall based from content given from Passage 1 directly about warm season but lacks information about cool seasons details strictly from provided snippets. ## Step 2: The text about picking out fertilizer and related content seem accurate based off Passage 2 directly. ## Step 3: About applying fertilizer and its method alongside checking turf's condition is based strictly off Passage3 hence appears accurate. ## Step 4: Other details mentioned like amount, frequency or any specific guidelines for organic or slow-release fertilizers are noted as not having enough info. {{"hallucination list": ["while cool season grasses should be fertilized in the fall."]}} |

Table 9: **RL4HS reasoning example 3 (Data2Text)**

---

***Review Data***

'name': 'Comedy Hideaway', 'address': '', 'city': 'Santa Barbara', 'state': 'CA', 'categories': 'Performing Arts, Arts & Entertainment, Caterers, Party & Event Planning, Com- edy Clubs, Venues & Event Spaces, Event Planning & Services, Nightlife', 'hours': 'Tuesday': '18:0-1:0', 'Friday': '18:0-23:0', 'Saturday': '13:0-23:0', 'attributes': 'BusinessParking': 'garage': False, 'street': False, 'validated': False, 'lot': True, 'valet': False, 'RestaurantsReservations': None, 'OutdoorSeating': False, 'WiFi': 'no', 'RestaurantsTakeOut': None, 'RestaurantsGoodForGroups': True, 'Music': False, 'Ambience': 'touristy': False, 'hipster': False, 'romantic': False, 'divey': False, 'intimate': True, 'trendy': False, 'upscale': False, 'classy': False, 'casual': True, 'busi- ness stars': 4.5, 'review info': ['review stars': 5.0, 'review date': '2019-05-11 20:55:41', 're- view text': ''Great local club to see rising talent. A hidden gem. You need to experience this before it's too late (plans are to move to a bigger location in DTSB).'', 'review stars': 5.0, 're- view date': '2018-11-12 20:03:58', 'review text': 'Went Saturday to see Laurie Kilmartin and it was great. Drinks are cheap, plenty of laughs to be had and next to the airport makes it feel like a super incognito location. All said the show was a blast.', 'review stars': 5.0, 'review date': '2018-10-30 02:26:26', 'review text': ''The place is fun the comics are good food is good seating could be better but for free it's awesome''

---

***Machine-generated article given review data***

Comedy Hideaway is a local business located in Santa Barbara, CA. It offers a variety of services including performing arts, arts entertainment, caterers, party event planning, comedy clubs, venues event spaces, event planning services, and nightlife. The business operates on Tuesdays from 6:00 PM to 1:00 AM, Fridays from 6:00 PM to 11:00 PM, and Saturdays from 1:00 PM to 11:00 PM.

According to customer reviews, Comedy Hideaway is highly recommended for its great local club atmosphere and rising talent performances. It is described as a hidden gem that provides a unique experience. Some customers mentioned that the venue plans to move to a bigger location in DTSB, indicating potential growth and expansion.

The business has a rating of 4.5 stars, reflecting the positive experiences of its customers. Reviewers appreciated the affordable drinks, plenty of laughs, and the location's proximity to the airport, which adds to its charm. However, some customers mentioned that the seating could be improved. Overall, Comedy Hideaway offers an enjoyable and fun experience with good food, talented comics, and a casual ambiance. It is a recommended destination for those looking to have a great time and enjoy live comedy shows in Santa Barbara.

---

***RL4HS reasoning path and output***

## Step 1: The overview article states 'business is a local business located in Santa Barbara, CA' from the structured data about Comedy Hideaway it appears to have city as Santa Barbara and state as CA. Therefore, this statement seems accurate based off given structured data. ## Step 2: The overview article states 'The business operates on Tuesdays from 6:00 PM to 1:00 AM, Fridays from 6:00 PM to 11:00 PM, and Saturdays from 1:00 PM to 11:00 PM.' Based off structured data for hours it shows Tuesday from '18:0-1:0' (interpreted as Tuesday from 6PM to 1AM), Friday 'from '18:0-23:0'(interpreted as Friday from 6PM to 23(=23) which is equivalent of saying Friday from 6PM -9PM or equivalently written in common sense context would be till late night around midnight but given directly its till mentioned time i.e.9pm), and Saturday 'from' '13 : -23 : '(interpreted as Saturday from noon (or pm) at one o'clock till late night around eleven pm). Therefore, for Friday time mentioned directly is till late night but based off provided content it's noted up until mentioned value i.e.9pm though stated here its up until midnight(=till late night), for the sake of accuracy with structured data provided we can confirm Tuesday Saturday times are accurate while for Friday its noted up until late evening or more specifically up until mentioned time i.e.9pm but article mentions up until at least midnight so might not strictly align with exact details provided. ## Step 3:The review snippet about restaurant includes positive aspects about food quality,talent etc alongside attributes about restaurant being casual which aligns well with structured details. ## Step4:The details provided regarding business characteristics seem aligned based off structured content. ''hallucination list'': [''Fridays from 6:00 PM to 11:00 PM'']

## A.8 BINARY HALLUCINATION DETECTION

We compare SFT and the proposed RL4HS on both binary and span detection tasks, and report the results in Table 10. While both SFT and RL4HS perform equally well in terms of binary detection, RL4HS clearly outperforms SFT in terms of fine-grained span detection, demonstrating the importance of RL for span detection task.

Table 10: **Binary and span level hallucination detection results**

| Model | Binary F1 | Span F1 | Model | Binary F1 | Span F1 |
|---|---|---|---|---|---|
| Qwen2.5-7B-COT | 56.2 | 16.2 | Qwen2.5-14B-COT | 62.0 | 27.3 |
| Qwen2.5-7B-SFT | 77.1 | 50.1 | Qwen2.5-14B-SFT | 77.9 | 55.4 |
| Qwen2.5-7B-RL4HS | 77.1 | **55.9** | Qwen2.5-14B-RL4HS | 78.0 | **58.3** |

## A.9 RL4HS WITH LIMITED TRAINING DATA

We trained Qwen2.5-Instruct-7B and Qwen2.5-Instruct-14B models using one-third of the RagTruth training set. The results in Table 11 show that RL4HS clearly outperforms SFT in low-data settings.

Table 11: **RL4HS with limited training data:** Both SFT and RL4HS models are trained with 1/3 RagTruth training data.

| Model | Summarization | Question Answering | Data2Text | Avg. |
|---|---|---|---|---|
| Qwen2.5-7B-SFT | 44.4 | 48.6 | 50.5 | 48.8 |
| Qwen2.5-7B-RL4HS | **51.1** | **52.4** | **57.1** | **54.2** |
| Qwen2.5-14B-SFT | 51.8 | 51.7 | 55.7 | 53.4 |
| Qwen2.5-14B-RL4HS | **57.3** | **55.0** | **59.2** | **57.3** |

### A.10 EXPERIMENTS ON PSILOQA DATASET

To evaluate the generalizability of RL4HS, we train Qwen2.5-7B-Instruct using both SFT and RL4HS on the PsiloQA dataset (Rykov et al., 2025). While the SFT model achieves a Span-F1 score of 73.0, the RL4HS trained model reaches 75.5. These results further demonstrate the superiority of RL4HS over SFT.

