# OpenReview forum: "Learning to Reason for Hallucination Span Detection"
_ICLR.cc/2026/Conference — ICLR 2026 Poster_

### Official Review · Reviewer_iny4 · 2025-10-16

**Soundness:** 2
**Presentation:** 2
**Contribution:** 2
**Rating:** 4
**Confidence:** 5

**Summary:**

The paper addresses hallucination span detection in large language models (LLMs), moving beyond binary detection to identify unsupported spans within generated text. The authors propose RL4HS, a reinforcement learning framework that leverages Chain-of-Thought (CoT) reasoning and introduces Class-Aware Policy Optimization (CAPO) to mitigate reward imbalance. Experiments on the RAGTruth benchmark (summarization, QA, data-to-text) show RL4HS outperforms both supervised fine-tuning and existing reasoning models, demonstrating the necessity of span-level rewards and in-domain reasoning for accurate hallucination detection.

**Strengths:**

- Novelty: First to train a reasoning-based hallucination span detector using RL with span-level rewards, addressing a gap in prior work focused on binary detection.
- Technical Contribution: CAPO effectively balances precision and recall, overcoming reward hacking issues found in standard GRPO.
- Empirical Rigor: Extensive experiments on RAGTruth across multiple tasks, withcomparisons to strong baselines (SFT, multi-view attention, proprietary reasoning models).
- Insightful Analysis: Ablation studies and case analysis convincingly show the benefits of in-domain reasoning and span-level reward optimization.

**Weaknesses:**

- Generality: Evaluation is limited to RAGTruth and a few CNLG tasks; broader applicability to other domains or real-world LLM outputs is not demonstrated.
- Model Scale: While RL4HS outperforms larger models, results for very large-scale proprietary models (e.g., GPT-5) are not fully explored.
- Complexity: The RL training setup (GRPO, CAPO) adds complexity and may be challenging to reproduce or deploy in production settings.
- Limited Error Analysis: The paper could benefit from deeper qualitative analysis of failure cases and limitations of RL4HS, especially in ambiguous or noisy contexts.
- Data Requirements: Reliance on span-level annotated data (RAGTruth) may limit adoption, as such datasets are rare.

**Questions:**

- Generalization: How does RL4HS perform on hallucination detection in domains outside RAGTruth (e.g., medical, legal, conversational AI)?
- Annotation Efficiency: Can RL4HS be adapted to settings with limited or noisy span-level annotations? Is weak supervision feasible?
- Deployment: What are the computational and practical challenges for deploying RL4HS in real-world LLM pipelines?
- Failure Modes: What types of hallucinations or contexts remain challenging for RL4HS? Any observed systematic errors?
- Comparison to Post-hoc Methods: How does RL4HS compare to post-hoc hallucination correction or filtering approaches in terms of accuracy and efficiency?

---

> ### Author Response · Authors · 2025-11-18
> **First version of rebuttal**
>
> We thank the reviewer for their valuable feedback and address their questions below:
>
> **Results for very large-scale proprietary models are not explored:**
> We already evaluated large-scale proprietary models (GPT-4o, GPT-5, GPT-o3). Please see Table. 1 in the original submission.
>
> **Applicability to real-world LLM outputs is not demonstrated**
> Please note that all the answers/summaries/overviews in RAGTruth dataset are generated by real-world LLMs, specifically
> GPT-3.5-turbo-0613, GPT-4-0613, Mistral-7b-Instruct, Llama-2-7B-chat, Llama-2-13B-chat and Llama-2-70B-chat. Please see the RAGTruth paper for details. So, all our results indeed demonstrate hallucination span detection performance for outputs generated by real-world LLMs.
>
> **Generalization:**
> Note that RAGTruth benchmark is not a single dataset. It consists of three distinct datasets from three different domains, and our experimental results in "unseen dataset" settings (Sec 5.3) clearly demonstrate the generalization ability of our RL4HS-trained models. Please see "common response to all reviewers" above for additional details (https://openreview.net/forum?id=ECAK3P92eg&noteId=9BkPZ6o7bi)
>
> **Deployment:**
> The computational requirements of RL4HS depend on the model size. As mentioned in Appendix A.3, our 7B/14B models were trained on a single 8-GPU node with 80GB GPUs. Given that many popular LLMs used today are trained using 100s of GPUs across several nodes, deploying RL4HS should be relatively easy. RL is increasingly becoming a core component of LLM training pipelines and there are open sourced libraries such as VERL (https://github.com/volcengine/verl) for efficient and production-ready RL training.
>
> **Comparison to Post-hoc Methods:**
> Note that this work focuses on detecting hallucinated content, which is different from correcting the detected hallucinations. Detection is a prerequisite step for correction. We cannot compare RL4HS with hallucination correction approaches since they focus on completely different tasks.
>
> **Annotation Efficiency:**
> We trained 7B and 14B models using SFT and RL4HS with only one-third of the original training data. The corresponding average span F1 scores are 48.8 (SFT) and 54.2 (RL4HS) for 7B model, and 53.4 (SFT) and 57.3 (RL4HS) for 14B model. RL4HS clearly outperforms SFT even in low-data settings.
>
> **Failure Modes:**
> We will share some examples soon.

---

> > ### Author Response · Authors · 2025-11-19
> > **Failure case examples**
> >
> > After going through several COTs, we couldn't concretely identify a set of systematic errors. However, one thing we have noticed in multiple samples is that the model gets confused when converting time from 24hr clock to 12hr clock. Below is one such example.
> > ****************
> > **Review Data**
> >
> > ’name’: ’Comedy Hideaway’, ’address’: ”, ’city’: ’Santa Barbara’, ’state’: ’CA’,
> > ’categories’: ’Performing Arts, Arts & Entertainment, Caterers, Party & Event Planning, Com-
> > edy Clubs, Venues & Event Spaces, Event Planning & Services, Nightlife’, ’hours’: ’Tuesday’:
> > ’18:0-1:0’, ’Friday’: ’18:0-23:0’, ’Saturday’: ’13:0-23:0’, ’attributes’: ’BusinessParking’: ’garage’:
> > False, ’street’: False, ’validated’: False, ’lot’: True, ’valet’: False, ’RestaurantsReservations’: None,
> > ’OutdoorSeating’: False, ’WiFi’: ’no’, ’RestaurantsTakeOut’: None, ’RestaurantsGoodForGroups’:
> > True, ’Music’: False, ’Ambience’: ’touristy’: False, ’hipster’: False, ’romantic’: False, ’divey’:
> > False, ’intimate’: True, ’trendy’: False, ’upscale’: False, ’classy’: False, ’casual’: True, ’busi-
> > ness stars’: 4.5, ’review info’: [’review stars’: 5.0, ’review date’: ’2019-05-11 20:55:41’, ’re-
> > view text’: ”Great local club to see rising talent. A hidden gem. You need to experience this
> > before it’s too late (plans are to move to a bigger location in DTSB).”, ’review stars’: 5.0, ’re-
> > view date’: ’2018-11-12 20:03:58’, ’review text’: ’Went Saturday to see Laurie Kilmartin and it
> > was great. Drinks are cheap, plenty of laughs to be had and next to the airport makes it feel like a
> > super incognito location. All said the show was a blast.’, ’review stars’: 5.0, ’review date’: ’2018-
> > 10-30 02:26:26’, ’review text’: ”The place is fun the comics are good food is good seating could be
> > better but for free it’s awesome”
> >
> > **Overview**
> >
> > Comedy Hideaway is a local business located in
> > Santa Barbara, CA. It offers a variety of services including performing arts, arts entertainment,
> > caterers, party event planning, comedy clubs, venues event spaces, event planning services, and
> > nightlife. The business operates on Tuesdays from 6:00 PM to 1:00 AM, Fridays from 6:00 PM to
> > 11:00 PM, and Saturdays from 1:00 PM to 11:00 PM.
> >
> > According to customer reviews, Comedy Hideaway is highly recommended for its great local club
> > atmosphere and rising talent performances. It is described as a hidden gem that provides a unique
> > experience. Some customers mentioned that the venue plans to move to a bigger location in DTSB,
> > indicating potential growth and expansion.
> >
> > The business has a rating of 4.5 stars, reflecting the positive experiences of its customers. Reviewers
> > appreciated the affordable drinks, plenty of laughs, and the location’s proximity to the airport, which
> > adds to its charm. However, some customers mentioned that the seating could be improved.
> >
> > Overall, Comedy Hideaway offers an enjoyable and fun experience with good food, talented comics,
> > and a casual ambiance. It is a recommended destination for those looking to have a great time and
> > enjoy live comedy shows in Santa Barbara.
> > ***********
> > **RL4HS Reasoning Path and prediction** ## Step 1: The overview article states ’business is a local business lo-
> > cated in Santa Barbara, CA’ from the structured data about Comedy Hideaway it appears to have city
> > as Santa Barbara and state as CA. Therefore, this statement seems accurate based off given struc-
> > tured data. ## Step 2: The overview article states ’The business operates on Tuesdays from 6:00 PM
> > to 1:00 AM, Fridays from 6:00 PM to 11:00 PM, and Saturdays from 1:00 PM to 11:00 PM.’ Based
> > off structured data for hours it shows Tuesday from ’18:0-1:0’ (interpreted as Tuesday from 6PM
> > to 1AM), Friday ’from ’18:0-23:0’(interpreted as Friday from 6PM to 23(=23) which is equivalent
> > of saying Friday from 6PM -9PM or equivalently written in common sense context would be till
> > late night around midnight but given directly its till mentioned time i.e.9pm), and Saturday ’from’
> > ’13 : -23 : ’(interpreted as Saturday from noon (or pm) at one o’clock till late night around eleven
> > pm). Therefore, for Friday time mentioned directly is till late night but based off provided content
> > it’s noted up until mentioned value i.e.9pm though stated here its up until midnight(=till late night),
> > for the sake of accuracy with structured data provided we can confirm Tuesday Saturday times are
> > accurate while for Friday its noted up until late evening or more specifically up until mentioned
> > time i.e.9pm but article mentions up until at least midnight so might not strictly align with exact
> > details provided. ## Step 3:The review snippet about restaurant includes positive aspects about food
> > quality,talent etc alongside attributes about restaurant being casual which aligns well with structured
> > details. ## Step4:The details provided regarding business characteristics seem aligned based off
> > structured content.
> >
> > ”hallucination list”: [”Fridays from 6:00 PM to 11:00 PM”]
> >
> > **Ground Truth**
> > No Hallucination

---

### Official Review · Reviewer_gYuN · 2025-10-30

**Soundness:** 3
**Presentation:** 3
**Contribution:** 2
**Rating:** 6
**Confidence:** 2

**Summary:**

The paper proposes RL4HS, a reinforcement learning framework for hallucination span detection in large language models. It builds upon Group Relative Policy Optimization (GRPO) and introduces Class-Aware Policy Optimization (CAPO) to handle class imbalance in rewards. The authors claim that RL4HS enables better reasoning for hallucination localization and outperforms supervised fine-tuning and prior reasoning-based baselines on the RAGTruth dataset. The study attempts to connect reasoning, reinforcement learning, and span-level detection but provides limited novelty beyond adapting existing RL techniques.

**Strengths:**

The topic—fine-grained hallucination detection—is relevant and timely, given the increasing importance of factual reliability in LLMs. The paper is well-organized, and the experimental setup is systematically described. The inclusion of span-level reward signals and the effort to analyze precision–recall imbalance show some awareness of practical issues in training reasoning-based detection models. The authors also provide qualitative examples to illustrate how reasoning might enhance model behavior.

**Weaknesses:**

Despite a clear structure, the work suffers from conceptual and methodological shallowness. The proposed RL4HS framework merely repackages existing GRPO methodology with a minor weighting adjustment, which can hardly be considered a significant algorithmic contribution. The claim that RL improves reasoning for hallucination span detection is weakly justified—there is no convincing evidence that the “reasoning” is genuinely learned rather than memorized through reward shaping. Experiments are conducted on a single dataset (RAGTruth), which raises concerns about generalizability. Furthermore, the comparison with GPT and Qwen models seems superficial and lacks control over model size, data exposure, and inference strategies. Many of the reported gains are marginal and could be attributed to overfitting or differences in fine-tuning procedures rather than the proposed RL method. The discussion of “in-domain reasoning” is vague and not theoretically supported. Overall, the paper feels more like an engineering report than a principled research contribution.

**Questions:**

What specific novelty does RL4HS offer over GRPO beyond the scaling factor (CAPO)? Why is this sufficient for publication in a top-tier conference?

How is “reasoning” objectively measured or verified in this work? Are the CoT traces evaluated for correctness or interpretability?

Given that results rely solely on RAGTruth, can the authors demonstrate performance on unseen domains or datasets?

How are baseline models such as GPT-5 or Qwen3 controlled for fair comparison—are they fine-tuned under identical conditions?

The reported F1 improvements seem small; can the authors provide statistical significance tests or ablations to confirm that these are not due to random variation?

Could similar gains be achieved simply with improved supervised training or calibration, without the complexity of reinforcement learning?

---

> ### Author Response · Authors · 2025-11-18
> **First version of rebuttal**
>
> We thank the reviewer for their valuable feedback and address their concerns below.
>
> **Contributions:**
> The main contributions of this work go beyond the introduction of CAPO.
> * To the best of our knowledge, this is the first work that advocates reasoning and RL for hallucination span detection. We provide clear motivation for reasoning and RL in Sec. 3.1 and demonstrate its effectiveness through thorough experimental validation in Sec. 5.
> * We demonstrate that the general reasoning ability learned by SOTA reasoning models based on data from various complex domains (such as math, coding, tool-use, etc) is not sufficient enough for hallucination span detection, and it is important to explicitly train the model for hallucination span detection using RL to equip it with the reasoning ability needed for this task. Our results clearly show that RL4HS-trained models outperform SOTA reasoning models by huge margin (see Table. 1).
>
> Given the increasing importance of factual reliability in LLMs, we believe that our contributions are significant since our findings and insights could provide valuable guidance to future research on this topic (also acknowledged by reviewer NqfQ).
>
> **Experiments on unseen datasets and domains:**
> Note that RAGTruth benchmark is not a single dataset. It consists of three distinct datasets from three different domains, and our experimental results in "unseen dataset" settings (Sec 5.3) clearly demonstrate the generalization ability of our RL4HS-trained models. Please see "common response to all reviewers" above for additional details (https://openreview.net/forum?id=ECAK3P92eg&noteId=9BkPZ6o7bi)
>
> **Finetuning baseline models:**
> Unfortunately, we cannot finetune proprietary GPT models since we do not have access to the model parameters. We use Qwen2.5 series 7B/14B models in our RL4HS experiments. For fair comparison, we also report results for same models with SFT in Table. 1. All models are evaluated using the same prompts (please see Appendix A.2)
>
> **Statistical significance:**
> We ran each experiment three times and report the average results in the main paper. Please refer to Appendix A.4 for the corresponding standard deviation values.
>
> **Do we need RL?**
> Results in Table. 1 clearly show that RL4HS outperforms SFT by a significant margin (5.8 points improvement in avg F1 score for 7B model 2.9 points improvement for 14B model). We believe that this improvement is due to the reasoning ability enhanced by RL training. Note that we only have hallucination span annotations without any COT traces, and hence cannot explicitly train a reasoning model using SFT.
>
> Figure. 1 in the original submission provides the motivation behind RL for span detection. It shows the SpanF1@K values with and without reasoning for different values of K. As K increases, the gap in terms of Span-F1@K increases significantly demonstrating the potential of CoT reasoning to generate at least one accurate prediction when sampled multiple times. These results provide clear motivation to use reinforcement learning for bringing the reasoning capacity of LLMs related to hallucination span detection to the forefront.
>
> **Is reasoning genuinely learned rather than memorized?**
> It is unclear what the reviewer meant by memorization here. We do not use any groundtruth COT traces during training for the model to memorize them. Also, all the performance numbers reported in the paper are computed on a held out test set which the model has not seen during training.
>
> Results in Table.1 clearly show that our reasoning model that has been trained specifically for hallucination span detection using RL outperforms several SOTA reasoning models by huge margins. Also, our RL-trained reasoning model outperforms the model trained on the exact same dataset using SFT. All these results clearly indicate that our RL training improves reasoning for hallucination span detection. Table. 3 illustrates the improved reasoning using a test example (not seen during).
>
> **Evaluating COT traces:**
> Upon manual inspection of several COT traces, we notice that the reasoning steps often involve identifying explicit claims in the output (i.e, summary or answer) and cross-checking these claims against the input context, which closely mimics how humans would do this task. We illustrate this using an example COT trace in Sec. 5.5. Note that exhaustively evaluating all the COT traces carefully is difficult since it involves significant amount of manual work.

---

> > ### Author Response · Authors · 2025-11-18
> > **First version of rebuttal (contd.)**
> >
> > **Discussion of “in-domain reasoning”:**
> > There are several reasoning models available today (e.g., Qwen3 series and GPT series) that have been trained on complex data domains such as math, coding, tool-calling, etc. However, the kind of reasoning steps a model performs when solving a math/coding problem could be different from the reasoning steps it needs to perform when detecting hallucination spans. We simply refer to "reasoning for hallucination span detection" as "in-domain reasoning" in this paper. Similarly, reasoning ability learned from training on other domains such as math, coding, tool-use, etc. can be interpreted as out-of-domain reasoning ability when our target task is hallucination span detection.
> >
> > We hypothesize that it is important to explicit train a model with hallucination span detection data to equip it with the reasoning needed for this task. To test this hypothesis (without over-customizing the reasoning ability to a specific dataset), we trained a reasoning model using hallucination span annotations in the context of two NLP tasks (e.g. summarization and QA) and evaluated the performance of the  model in the context of a different NLP task (e.g. Data-to-text). Figure. 5 in the paper shows the corresponding results. On average, our trained models RL4HS-OOD-7B clearly outperform other reasoning-based models such as Qwen3-7B, Qwen3-14B, GPT-4o-mini w/ COT verifying our hypothesis.

---

### Official Review · Reviewer_mScM · 2025-11-01

**Soundness:** 4
**Presentation:** 4
**Contribution:** 4
**Rating:** 8
**Confidence:** 5

**Summary:**

The authors reframe hallucination detection as a reinforcement learning problem instead of a simple classification one. The authors first show that using Chain-of-Thought (CoT) reasoning before predicting hallucinated spans makes the model’s predictions more diverse — so when decoding multiple times (say, K attempts), at least one prediction tends to be correct. Building on this, they design RL4HS, which fine-tunes a model using Group Relative Policy Optimization (GRPO) with a span-level F1 reward to explicitly encourage reasoning that improves hallucination span localization. However, they find that if the input has no hallucination, directly giving reward = 1.0 biases the model toward always predicting “no hallucination.” To fix this, they propose Class-Aware Policy Optimization (CAPO), which scales the non-hallucination advantages (by 0.5) to balance rewards between classes. They also confirm that simple fixes like Dr.GRPO cannot solve this imbalance. Experiments on RAGTruth (covering summarization, QA, and data-to-text) show that RL4HS significantly outperforms supervised fine-tuning and existing reasoning models, proving that reinforcement learning with span-level rewards and in-domain reasoning is essential for robust hallucination detection

**Strengths:**

- The paper offers a novel reformulation of hallucination span detection as a reinforcement learning problem, which is conceptually original and well-motivated.
- It carefully designs a span-level reward and introduces class-aware scaling to prevent reward hacking, showing thoughtful methodological innovation.
- The motivation analysis with Span-F1@K clearly illustrates the benefit of reasoning diversity and provides strong empirical grounding.
- Experimental results demonstrate substantial improvements over both supervised and reasoning baselines, highlighting the method’s effectiveness and significance.
- The paper is clearly written and logically structured, making complex ideas easy to follow.

**Weaknesses:**

- The approach mainly focuses on the RAGTruth benchmark; it remains unclear how well it generalizes to other OOD data. But indeed the author shows good transferability among different three subsets under RAGTruth, using and holdout setting.
- It could be beneficial to analyze and categorize what kinds of strategy the RL4HS model uses for producing more accurate hallucination span detection, by some human evaluation on the CoT paths.
- Here are only one qualitative study examples shown in the paper. It would be good if the authors can provide more in the appendix.

**Questions:**

N/A

---

> ### Author Response · Authors · 2025-11-18
> **Thank you for your positive feedback.**
>
> We thank the reviewer for clearly highlighting the strengths of our work and address their comments below:
>
> **Generalization:**
> Note that RAGTruth benchmark is not a single dataset. It consists of three distinct datasets from three different domains, and our experimental results in "unseen dataset" settings (Sec 5.3) clearly demonstrate the generalization ability of our RL4HS-trained models. Please see "common response to all reviewers" above for additional details (https://openreview.net/forum?id=ECAK3P92eg&noteId=9BkPZ6o7bi)
>
> **Analyzing COTs:**
> Upon manual inspection of several COT traces, we notice that the reasoning steps often involve identifying explicit claims in the output (i.e, article or answer) and cross-checking these claims against the input context, which closely mimics how humans would do this task. We illustrate this using an example COT trace in Sec. 5.5. We will add more examples in the final version of the paper. Note that exhaustively evaluating all the COT traces carefully is difficult since it involves significant amount of manual work.

---

### Official Review · Reviewer_NqfQ · 2025-11-01

**Soundness:** 2
**Presentation:** 3
**Contribution:** 2
**Rating:** 4
**Confidence:** 4

**Summary:**

This paper propose RL4HS, a reinforcement learning framework that designed for hallucination span detection. The authors first conduct experiments comparing the pass@k performance, demonstrating that incorporating reasoning can be beneficial for span detection. Then, they train the model using GRPO with a span-level reward function. To address reward hacking issues caused by the imbalanced advantages between hallucination and non-hallucination cases, this work further introduce Class-Aware Policy Optimization (CAPO), which adjusts the advantages for non-hallucination predictions.

**Strengths:**

1. The paper presents a systematic scaling analysis, highlighting the potential of reasoning in improving hallucination span detection. These findings and insights could be valuable for guiding future research.

2. The study clearly identifies and analyzes the reward hacking issue caused by the imbalanced reward designs, and the proposed CAPO method offers a effective solutions.

**Weaknesses:**

1. Experiments are conducted exclusively on RAGTruth. It is unclear whether the proposed method generalizes to other hallucination datasets. There are other hallucination detection benchmarks with span-level annotations, such as FAVA, that could be included for a more comprehensive evaluation.

2. Although the authors compare to several reasoning and proprietary models, hallucination-specific baselines is limited.

3. The paper does not report case-level (or binary-level) results, so it remains unclear whether the span-level reward leads to consistent gains in overall performance.

4. The SFT baseline seems suboptimal relative to the original RAGTruth paper, possibly due to inappropriate learning rates (1e-6 in appendix) or hyperparameters, which may underestimate supervised performance.

**Questions:**

See above section

---

> ### Author Response · Authors · 2025-11-18
> **First version of rebuttal**
>
> We thank the reviewer for their valuable feedback and address their concerns below.
>
> **Hallucination specific baselines:**
> While there are numerous existing works on binary hallucination detection, the literature on span detection is relatively limited. Methods designed for binary detection do not typically generate hallucination span predictions. The original RAGTruth paper reports span detection results for two approaches: (1) Prompting off-the-shelf LLMs, and (2) SFT with span-level annotations. Another recent work [1] also uses SFT as the baseline. We report results for both these baselines in our work. Table 1 (Sec. 5), shows prompting-based results for several SOTA reasoning and non-reasoning models and SFT results for Qwen-2.5 models which have been used in RL4HS experiments.
>
> [1] Hallucinated Span Detection with Multi-View Attention Features, CoRR, abs:/2504.04335.
>
>
> **Binary detection results:**
> Below table shows both binary detection and span prediction results. While both SFT and RL4HS perform equally well in terms of binary prediction, RL4HS clearly outperforms SFT in terms of fine-grained span prediction. These results further demonstrate the importance of RL for span prediction.
>
> | Model                        | Binary F1  |  Span F1 | Model                        | Binary F1  |  Span F1 |
> | --------                       | -----------  | --------    | --------                       | -----------  | --------    |
> | Qwen2.5-7B-COT     |       56.2     |   16.2      | Qwen2.5-14B-COT     |       62.0  |   27.3       |
> | Qwen2.5-7B-SFT      |      77.1      |   50.1      | Qwen2.5-14B-SFT      |      77.9   |   55.4      |
> | Qwen2.5-7B-RL4HS |      77.1      |  55.9       | Qwen2.5-14B-RL4HS |      78.0   |  58.3       |
>
>
> **SFT baseline seems suboptimal relative to original RAGTruth paper:**
> On average, our SFT results are actually better than the SFT results reported in the original RAGTruth paper. SFT results in the original RAGTruth paper can be found in Table. 6 of https://arxiv.org/pdf/2401.00396
>
> |    SFT        |  Model            | Summ | QA   | D2T | Avg  |
> | ----------   | ---------------  | -------  | ----- | ----  | ----- |
> | Ours         | Qwen2.5-14B | 52.7    | 53.9 | 59.6 | **55.4** |
> | RAGTruth | Llama2-13B    | 38.8    | 58.2 | 53.5 | 52.7 |
>
>
> **Generalization**
> Note that RAGTruth benchmark is not a single dataset. It consists of three distinct datasets from three different domains, and our experimental results in "unseen dataset" settings (Sec 5.3) clearly demonstrate the generalization ability of our RL4HS-trained models. Please see "common response to all reviewers" above for additional details (https://openreview.net/forum?id=ECAK3P92eg&noteId=9BkPZ6o7bi)

---

### Author Response · Authors · 2025-11-18
**Common response to all reviewers - Generalization performance**

We thank all the reviewers for their valuable feedback. In this comment, we address the questions regarding generalization raised by multiple reviewers.

We would like to bring to the attention of the reviewers that RAGTruth benchmark is not a single dataset. It consists of three different datasets sourced from three different sources focusing on three different tasks:
* Summarization dataset focuses on news summarization and is sourced from CNN/DM dataset [1]
* Question answering dataset focuses on RAG-based QA and is sourced from MS MARCO dataset [2]
* Data-to-text dataset focuses on generating overviews of structured data and is sourced from Yelp Open Dataset [3].

We conducted experiments by training our model on two of the above three datasets and testing it on the remaining one. We did these experiments for all three possible combinations (of training on two datasets and testing on one) and reported the results in Sec 5.3 of the original submission. Our trained models (RL4HS-OOD-7B)  outperform other reasoning-based models such as Qwen3-7B, Qwen3-14B, GPT-4o-mini w/ COT demonstrating the generalization of the reasoning ability learned by models trained with RL4HS.

For the rebuttal, we also ran SFT experiments in these OOD settings, and the SFT-trained models perform significantly worse (32.4 vs 37.2 average span F1 score) when compared to our RL4HS trained models.

[1] Get to the point: Summarization with pointer-generator networks, ACL 2017

[2] MS MARCO: A Human Generated MAchine Reading COmprehension Dataset, CoCo@NeurIPS 2016.

[3] Yelp Open Dataset (https://business.yelp.com/data/resources/open-dataset/), 2021.

---

> ### Author Response · Authors · 2025-11-18
> **Common response to all reviewers - Additional COT examples**
>
> Here, we provide an example of the reasoning trace generated by our RL4HS trained model when detecting hallucination spans in a sample from the Question-Answering dataset.
>
> *****************
>
> **Question**
> differences and similarities between red and white blood cells
>
> **Retrieved Passages**
>
> **passage 1:**
> Shares \& Saves. Save. Blood is made up of three parts: red cells, white cells and platelets. Red blood cells are responsible for the oxygen delivery from the lungs to the tissue in the body. White blood cells are a part of the immune system. These components circulate in the fluids in the blood vessels, which is pumped by the heart.hares \& Saves. Save. Blood is made up of three parts: red cells, white cells and platelets. Red blood cells are responsible for the oxygen delivery from the lungs to the tissue in the body. White blood cells are a part of the immune system.
>
> **passage 2:** 1. Both red blood cells and white blood cells are found in blood. 2. Both red blood cells and white blood cells can be carried by blood from one part to other part of our body … .3.The red blood cells and the White Blood Cells can are found in the circulatory system. 4.It both have cell membrane and cytoplasm.hite blood cells are part of the Immune System and help the body combat disease, viruses, etc. Red blood cells are part of the Circulatory System and carry things such as oxy … gen throughout the body.
>
> **passage 3:**
> Best Answer: Similarities: - RBCs, WBCs are formed elements of blood. - RBCs, WBCs are formed from pluripotent stem cells in red bone marrow.est Answer: Similarities: - RBCs, WBCs are formed elements of blood. - RBCs, WBCs are formed from pluripotent stem cells in red bone marrow.
>
> **Answer**
>
>  Based on the given passages, here are the differences and similarities between red and white blood cells:
>
> Differences:
>
> * Function: Red blood cells are responsible for delivering oxygen from the lungs to the body's tissues, while white blood cells are part of the immune system and help combat diseases and infections.
> * Structure: Red blood cells have a unique biconcave disk shape that allows them to maximize their surface area for gas exchange, whereas white blood cells have a irregular, amoeba-like shape.
>
> Similarities:
>
> * Formation: Both red and white blood cells are formed from pluripotent stem cells in the red bone marrow.
> * Presence in circulatory system: Both red and white blood cells are found in the circulatory system and can be carried by blood from one part of the body to another.
> * Cellular components: Both red and white blood cells have a cell membrane and cytoplasm.
>
> Therefore, while red and white blood cells have some similarities in terms of their formation and presence in the circulatory system, they differ in their function and structure.
>
> ***************
>
> **RL4HS Reasoning Path and output**
>
>  \#\# Step 1: The statement 'Differences:' with '* Function: Red blood cells are responsible for delivering oxygen from the lungs to the body's tissues, while white blood cells are part of the immune system and help combat diseases and infections.' based from Passage 1 and Passage 2 seems accurate from provided content about their roles directly.
> \#\# Step 2: The statement '* Structure: Red blood cells have a unique biconcave disk shape that allows them to maximize their surface area for gas exchange, whereas white blood cells have a irregular, amoeba-like shape.' appears without strong basis from any of given related passages (Passage1, Passage2 or Passage3). None of these text snippets confirm details about structure or shapes directly.
> \#\# Step 3: The statement 'Similarities:' with content about formation based off passage3 seem accurate. Details provided from snippet directly.
> \#\# Step 4: The statement 'Presence in circulatory system:' along with 'Cellular components:' seem based off passage2 accurately.
>
> {{"hallucination list": ["* Structure: Red blood cells have a unique biconcave disk shape that allows them to maximize their surface area for gas exchange, whereas white blood cells have a irregular, amoeba-like shape."]}}

---

> > ### Author Response · Authors · 2025-11-18
> > **Common response to all reviewers - Additional COT examples**
> >
> > Here, we provide one more example of the reasoning trace generated by our RL4HS trained model.
> > ***********
> > **Question**
> > what is the proper way to fertilize your grass
> >
> > **Retrieved Passage**
> >
> > **passage 1:**
> > Fertilize at the right time of year. In order to your fertilizer to work best, you'll need to time it so that you fertilize at the beginning of the growing season. This is different for different types of grasses. Fertilize warm season grasses when the grass begins to turn green at the beginning of spring.
> >
> > **passage 2:**
> > Picking Out Fertilizer. 1  1. Know your grass. The type of grass you have will determine what type of fertilizer you need to use, and how often you need to fertilize. 2  2. Do a pH soil test. This measures the alkalinity or acidity of the soil, and helps to determine what pH your fertilizer should have. 3  3. Measure your lawn.
> >
> > **passage 3:**
> > Pour in the rest of the fertilizer. Apply the rest of fertilizer in a perpendicular pattern. Step on the turf and if the leaf blade bounces back up, your lawn is in good shape. If it lays flat, you've over-fertilized.
> >
> > **Answer**
> >
> > Based on the provided passages, the proper way to fertilize your grass is as follows:
> >
> > 1. Time your fertilization at the beginning of the growing season, which varies for different types of grasses. Warm season grasses should be fertilized when the grass turns green in the early spring, while cool season grasses should be fertilized in the fall. (Passage 1)
> > 2. Choose a fertilizer that is appropriate for your specific type of grass and perform a pH soil test to determine the correct pH level for your fertilizer. (Passage 2)
> > 3. Apply the remaining fertilizer in a perpendicular pattern, and check if the leaf blade bounces back up after stepping on the turf to ensure that your lawn is in good shape. If the leaf blade lies flat, you may have over-fertilized. (Passage 3)
> >
> > Unable to answer based on given passages. The passages do not provide information on the amount of fertilizer to use, how often to fertilize, or any specific guidelines for organic or slow-release fertilizers.
> >
> > **********
> > **RL4HS Reasoning Path and output**
> >
> > \#\# Step 1: The statement '1. Time your fertilization at the beginning of the growing season, which varies for different types of grasses. Warm season grasses should be fertilized when the grass turns green in the early spring, while cool season grasses should be fertilized in the fall. (Passage 1)' from answer seems to include about cool season grasses part about being fertilized in fall based from content given from Passage 1 directly about warm season but lacks information about cool seasons details strictly from provided snippets.
> > \#\# Step 2: The text about picking out fertilizer and related content seem accurate based off Passage 2 directly.
> > \#\# Step 3: About applying fertilizer and its method alongside checking turf's condition is based strictly off Passage3 hence appears accurate.
> > \#\# Step 4: Other details mentioned like amount, frequency or any specific guidelines for organic or slow-release fertilizers are noted as not having enough info.
> >
> > {{"hallucination list": ["while cool season grasses should be fertilized in the fall."]}}

---

> ### Author Response · Authors · 2025-11-26
> **Common Response to all reviewers - Additional experiments on new PsiloQA dataset**
>
> We trained Qwen2.5-7B model using SFT and RL4HS on recently released PsiloQA dataset [1]. While SFT model achieves Span-F1 score of 73.0, RL4HS trained model achieves 75.5 clearly demonstrating the superiority of RL4HS over SFT.
>
> [1] "When Models Lie, We Learn: Multilingual Span-Level Hallucination Detection with PsiloQA", EMNLP Findings 2025.

---

### Author Response · Authors · 2025-12-04
**General Response: Summary of Strengths**

We are grateful for the thoughtful and constructive feedback provided on our paper “Learning to Reason for Hallucination Span Detection”. Below, we summarize the main strengths highlighted by the reviewers in the paper. We sincerely appreciate the reviewers’ detailed and constructive input. The feedback has substantially strengthened our manuscript, both in technical detail and clarity.
We believe the rebuttal and additional experiments meaningfully address the raised concerns and further highlight the significance of RL-enhanced in-domain reasoning for hallucination span detection.

### Summary of Strengths:
1. **Novelty \& Contribution**
(Reviewers mScM, iny4):
Reviewers acknowledged that our work is the first to advocate for explicit reasoning and reinforcement learning for hallucination span detection, moving beyond prior binary detection frameworks.
Our RL4HS framework introduces span-level rewards and CAPO to mitigate reward imbalance, demonstrating a meaningful methodological advance.

2. **Importance of the Study Topic**
(Reviewers gYuN, iny4, mScM):
Multiple reviewers emphasized the growing importance of fine-grained hallucination detection as LLMs are deployed in high-stakes applications. Our framing of hallucination span detection as a multi-step decision-making and reasoning task addresses a critical reliability challenge in modern LLM systems.

3. **Well-Motivated Approach**
(Reviewers NqfQ, mScM):
Reviewers found the motivation for RL4HS clearly articulated. The analysis of span-F1@K demonstrates why reasoning diversity is beneficial, and the paper provides a compelling argument for why off-the-shelf reasoning models are insufficient and why learning task-specific reasoning via RL is necessary for accurate hallucination span localization.


4. **Well-Written and Clearly Presented**
(Reviewers gYuN, iny4, mScM):
The paper was praised for its clarity and organization. Reviewers noted that the methodology, reward design, and experimental setup were easy to follow, and the writing made complex ideas accessible. The inclusion of detailed examples contributed to the readability and clarity of the work.

5. **Insightful Analysis**
(Reviewers NqfQ, iny4, mScM):
Reviewers highlighted the paper’s thorough and insightful analysis, noting that it clearly diagnoses the reward-imbalance issue and demonstrates how CAPO effectively addresses it. The ablation studies, cross-domain generalization experiments across the three RAGTruth tasks, and qualitative reasoning traces collectively provide strong evidence for how RL4HS enhances in-domain reasoning for hallucination span detection, offering valuable guidance for future work in this area.

6. **Strong Experimental Result** :
    - Our work provides extensive and rigorous empirical evaluation demonstrating the effectiveness and robustness of RL4HS. The **Span-F1@K** motivation analysis clearly shows the benefit of reasoning diversity for hallucination span detection (refer to Sec. 3.1 and Fig. 1).

     - Across all three RAGTruth datasets, RL4HS consistently outperforms both supervised fine-tuning and strong reasoning baselines, including larger proprietary models (refer to Sec. 5.1 and Table. 1). The new results obtained during rebuttal phase on PsiloQA dataset also clearly demonstrate the superiority of **RL4HS (75.5 Span-F1) over SFT (73.0 Span- F1)** (Please see the rebuttal response on `Generalization performance` ).

      - RL4HS outperforms various off-the-shelf reasoning models such as Qwen3-7B, Qwen3-14B, GPT-4o-mini w/ COT in **out-of-domain evaluation settings** (refer to Sec 5.3 Fig. 5) demonstrating the generalization of the reasoning ability learned by models trained with RL4HS.
    The new results obtained during rebuttal phase, in OOD settings, RL4HS also outperforms SFT trained models (**RL4HS 37.2 v.s SFT 32.4** average Span-F1 score, please see the rebuttal response on `Generalization performance`).

    - Moreover, RL4HS remains effective even with less training data. In a new experiment added during the rebuttal, we trained 7B and 14B models using only **one-third of the original data**. The 7B model achieved **54.2 Span-F1 with RL4HS vs. 48.8 with SFT**, and the 14B model reached **57.3 vs. 53.4**. This shows that RL4HS clearly outperforms SFT even in low-data settings. (see the rebuttal response to `reviewer iny4`).

---

### Meta-Review · Area_Chair_emre · 2026-01-06

**Summary:**

This paper proposes RL4HS for hallucination span detection, framing span localization as a sequential decision problem and training with a modified GRPO using a span-level reward. The paper introduces CAPO, a weighted class-aware adjustment of GRPO intended to mitigate reward imbalance for no-hallucination cases.

Reviewers agree the task is important and the paper is clearly written, and the results on RAGTruth are competitive. However, there is substantial disagreement on whether the contribution goes beyond an engineering adaptation of existing RL algorithm. The rebuttal does not convincingly explain why CAPO is algorithmically superior to GRPO. Beyond stating that CAPO rescales advantages to address class imbalance, the authors do not analyze GRPO’s failure mode, justify CAPO from an optimization or learning-dynamics perspective, or rule out simpler alternatives such as reward reweighting or calibrated supervised training. Consequently, the novelty concern regarding CAPO remains unaddressed.

**Reviewer Concerns:**

Addressed or partially addressed by the rebuttal:
- Lack of case-level or binary-level reporting: The authors added binary detection results alongside span-level results and showed that RL4HS improves span F1 while keeping binary F1 essentially unchanged. This directly addresses the concern that span-level optimization might not translate to consistent case-level behavior.

- Potentially suboptimal SFT baseline: The rebuttal provided a comparison against the original RAGTruth paper’s reported SFT numbers and argued their SFT is competitive or stronger on average. While cross-paper comparisons are imperfect due to different backbones, the response partially alleviates the concern that gains come purely from an under-tuned SFT baseline.

- Generalization beyond a single dataset: The authors emphasized that RAGTruth includes three tasks/domains and provided leave-one-task-out evaluation, plus an additional experiment on PsiloQA during rebuttal. This strengthens the empirical story, but it still leaves open whether the method generalizes to substantially different benchmarks or distributions.

Outstanding / not convincingly addressed:
- External dataset coverage and breadth: The rebuttal does not add evaluation on other established span-level benchmarks (e.g., FAVA) and largely reframes “generalization” within RAGTruth’s internal splits. The added PsiloQA result helps, but broader coverage remains limited.

- Novelty and necessity of CAPO relative to GRPO (raised strongly by Reviewer gYuN and implicitly by others): While the rebuttal claims CAPO mitigates reward imbalance, it does not provide a sufficiently crisp explanation of why GRPO fails in this setting, nor a principled justification that CAPO is more than a heuristic scaling. Clean ablations isolating GRPO vs GRPO+CAPO (and comparisons to simpler fixes like reward reweighting or calibrated SFT) are still missing, so concerns about incremental novelty remain.

**Reviewer Scores:**

Reviewer NqfQ (4)
Likely unchanged (4 → 4). Added binary results and SFT clarification help, but core concerns about benchmark breadth and generalization remain.

Reviewer mScM (8)
Score appears generous and should be viewed as an enthusiastic outlier.

Reviewer gYuN (6)
Likely unchanged (6 → 6). Core concerns about limited novelty beyond CAPO and weak evidence for learned reasoning are not addressed by the rebuttal.

Reviewer iny4 (4)
Likely unchanged (4 → 4). Clarifications were provided, but concerns about generality, complexity, and limited failure analysis remain.

---

### Decision · Program_Chairs · 2026-01-26

Accept (Poster)